# The quantity of CD40 signaling determines the differentiation of B cells into functionally distinct memory cell subsets

Takuya Koike, Koshi Harada, Shu Horiuchi, Daisuke Kitamura*

Division of Molecular Biology, Research Institute for Biomedical Sciences (RIBS), Tokyo University of Science, Noda, Japan

**Abstract** In mice, memory B ($B_{mem}$) cells can be divided into two subpopulations: $CD80^{hi}$ $B_{mem}$ cells, which preferentially differentiate into plasma cells; and $CD80^{lo}$ $B_{mem}$ cells, which become germinal center (GC) B cells during a recall response. We demonstrate that these distinct responses can be B-cell-intrinsic and essentially independent of B-cell receptor (BCR) isotypes. Furthermore, we find that the development of $CD80^{hi}$ $B_{mem}$ cells in the primary immune response requires follicular helper T cells, a relatively strong CD40 signal and a high-affinity BCR on B cells, whereas the development of $CD80^{lo}$ $B_{mem}$ cells does not. Quantitative differences in CD40 stimulation were enough to recapitulate the distinct B cell fate decisions in an in vitro culture system. The quantity of CD40 signaling appears to be translated into NF-κB activation, followed by BATF upregulation that promotes $B_{mem}$ cell differentiation from GC B cells.

DOI: https://doi.org/10.7554/eLife.44245.001

## Introduction

Memory B ($B_{mem}$) cells are crucial for humoral immunity, preventing the spread of re-infecting viruses and bacteria by rapidly producing large amounts of class-switched antibodies (Abs) against these pathogens. $B_{mem}$ cells also regenerate themselves with improved affinity for their cognate antigen through the germinal center (GC) reaction upon each iterative infection. The rapid and exaggerated response of $B_{mem}$ cells has been attributed to their B-cell receptor (BCR) isotype, namely membrane-bound IgG (mIgG). The relatively long cytoplasmic tail of mIgG, as compared to that of membrane-bound IgM (mIgM) which consists of only three amino acids, contains specific motifs that recruit signaling molecules such as Grb2 and SAP97 after BCR cross-linking; these motifs and molecules are required for enhanced BCR signaling and plasma cell (PC) formation upon rechallenge (*Engels et al., 2009*; *Kaisho et al., 1997*; *Liu et al., 2012*; *Lutz et al., 2015*). At odds with this model is the finding that naïve B cells expressing NP-specific mIgG1, which were derived from cloned mice generated from a single IgG1+ $B_{mem}$ cell, expanded to a similar extent as NP-specific (IgH-knock-in) IgM+ naïve B cells, and both B cell types predominantly differentiated into GC B cells rather than PCs upon primary immunization (*Kometani et al., 2013*). These observations suggest that the heightened capacity of $B_{mem}$ cells to differentiate into PCs cannot be attributed solely to the expression of mIgG1, but also depends on some cell-intrinsic status, such as a reduced expression of Bach2, a transcription factor that suppresses PC differentiation (*He et al., 2017*; *Kometani et al., 2013*). In support of this notion, human $B_{mem}$ cells differentiate into PCs better than do naïve B cells in antigen-free in vitro cultures (*Arpin et al., 1997*).

As a further refinement in our understanding of the functional properties of $B_{mem}$ cells, it has recently been proposed that the $B_{mem}$ cell pool can be divided into distinct subsets on the basis of

*For correspondence:
kitamura@rs.noda.tus.ac.jp

Competing interests: The authors declare that no competing interests exist.

their potential to generate PCs or GC B cells upon antigen encounter. Earlier reports indicated that $B_{mem}$ cells consist of mIgG$^+$ cells and mIgM$^+$ cells, with the former prone to becoming PCs and the latter GC B cells (*Dogan et al., 2009*; *Pape et al., 2011*), although a recent report suggests a more complex scenario (*McHeyzer-Williams et al., 2015*). Another study proposed that functionally different subsets can be phenotypically defined using surface markers, CD80 and PD-L2: CD80$^+$ PD-L2$^+$ $B_{mem}$ cells are prone to differentiate into PCs, whereas CD80$^-$ PD-L2$^-$ $B_{mem}$ cells preferentially form GC upon secondary immunization, and CD80$^-$ PD-L2$^+$ $B_{mem}$ cells are intermediate between the two but functionally closer to the CD80$^-$ PD-L2$^-$ $B_{mem}$ cells (*Zuccarino-Catania et al., 2014*). Consistent with the BCR-isotype-based classification, the majority of IgG1$^+$ $B_{mem}$ cells had the CD80$^+$ PD-L2$^+$ phenotype, whereas IgM$^+$ $B_{mem}$ cells were dominated by CD80$^-$ PD-L2$^-$ cells. In addition, CD73 has been used to further dissect the PC-prone subpopulation as CD80$^+$ PD-L2$^+$ CD73$^+$ $B_{mem}$ cells (*He et al., 2017*).

Circulating memory T cells have also been functionally divided into two subsets termed effector memory T ($T_{EM}$) and central memory T ($T_{CM}$) cells. $T_{EM}$ cells, which lack the lymph node homing receptors CD62L and CCR7, produce abundant effector cytokines and cytotoxic granules, whereas $T_{CM}$ cells expressing both CD62L and CCR7 have a greater potential for proliferation and self-renewal (*Mueller et al., 2013*). The PC-prone $B_{mem}$ cells may resemble the $T_{EM}$ cells in terms of their effector function, whereas the GC-B cell-prone $B_{mem}$ cells are similar to the self-renewable $T_{CM}$ cells.

At present, it is not clear how these two subsets diverge during the primary immune response. It has been reported that CD80$^+$ PD-L2$^+$ $B_{mem}$ cells have the highest affinity for antigen, whereas the double negative cells have the lowest affinity (*Tomayko et al., 2010*; *Zuccarino-Catania et al., 2014*). Moreover, it is generally accepted that B cells expressing BCR of higher affinity are prone to differentiate into PCs and those of lower affinity into GC B cells (*Ochiai et al., 2013*; *Paus et al., 2006*; *Sciammas et al., 2011*). Together with the analysis of the frequency of BCR somatic mutations and the timing of $B_{mem}$ cell generation (*Weisel et al., 2016*), it has been suggested that PC-prone $B_{mem}$ cells, including IgG1$^+$ and CD80$^+$ PD-L2$^+$ $B_{mem}$ cells, are mainly derived from the GC, whereas most of GC-B cell-prone $B_{mem}$ cells, including IgM$^+$ and CD80$^-$ PD-L2$^-$ $B_{mem}$ cells, are generated before GC formation. It remains unclear, however, which signals determine the distinct $B_{mem}$ cell fates. Thus, we sought to define the signaling mechanism for the bifurcated generation of $B_{mem}$ cells.

## Results

### Characterization of CD80$^{hi}$ and CD80$^{lo}$ memory B cells

On the basis of previous reports showing that IgG1$^+$ $B_{mem}$ cells are mainly composed of CD80$^+$ PD-L2$^+$ and CD80$^-$ PD-L2$^+$ $B_{mem}$ cells and that CD80$^-$ PD-L2$^+$ and CD80$^-$ PD-L2$^-$ $B_{mem}$ cells are functionally similar (*He et al., 2017*; *Zuccarino-Catania et al., 2014*), we hypothesized that the proposed $B_{mem}$ cell subsets could be distinguished simply by the expression of CD80, as CD80$^{hi}$ or CD80$^{lo}$ $B_{mem}$ cells. Consistent with these reports, most of the CD80$^{hi}$ $B_{mem}$ cells expressed PD-L2 and CD73, thus constituting the reported affinity-matured subset (*He et al., 2017*; *Tomayko et al., 2010*; *Zuccarino-Catania et al., 2014*), whereas the CD80$^{lo}$ $B_{mem}$ cells consisted of PD-L2$^+$ and PD-L2$^-$, as well as CD73$^+$ and CD73$^-$ subpopulations (*Figure 1a*). Although CD62L was reported to be expressed on $B_{mem}$ cells (*Anderson et al., 2007*), the majority of CD80$^{hi}$ $B_{mem}$ cells express a lower level of CD62L (a phenotype that somewhat resembles $T_{EM}$ cells), whereas most CD80$^{lo}$ $B_{mem}$ cells express a higher level of CD62L (a phenotype that somewhat resembles $T_{CM}$ cells). Both CD80$^{hi}$ and CD80$^{lo}$ $B_{mem}$ cells expressed FAS (also called CD95 or APO1) at a higher level than naïve B cells, as previously reported (*Anderson et al., 2007*), but at a lower level than GC B cells, and, as expected, they were GL7$^-$.

In order to examine in vitro whether the CD80$^{hi}$ and CD80$^{lo}$ $B_{mem}$ cells are intrinsically biased in their differentiation fate toward PCs or GC B cells, we transferred into B6 mice allotypically marked (CD45.1$^+$) B cells of B1-8 knock-in (ki) mice, whose knock-in IgH chain, when combined with the λL chain, forms an NP-specific BCR, and immunized these mice with NP-CGG. From these mice, we sorted CD80$^{hi}$ and CD80$^{lo}$ $B_{mem}$ cells, either IgG1$^+$ or IgG1$^-$, and cultured them with IL-21 on feeder cells that express exogenous CD40L and BAFF (40LB) (*Nojima et al., 2011*; *Takatsuka et al., 2018*). Under these conditions, CD80$^{hi}$ $B_{mem}$ cells differentiated more preferentially into CD138$^+$

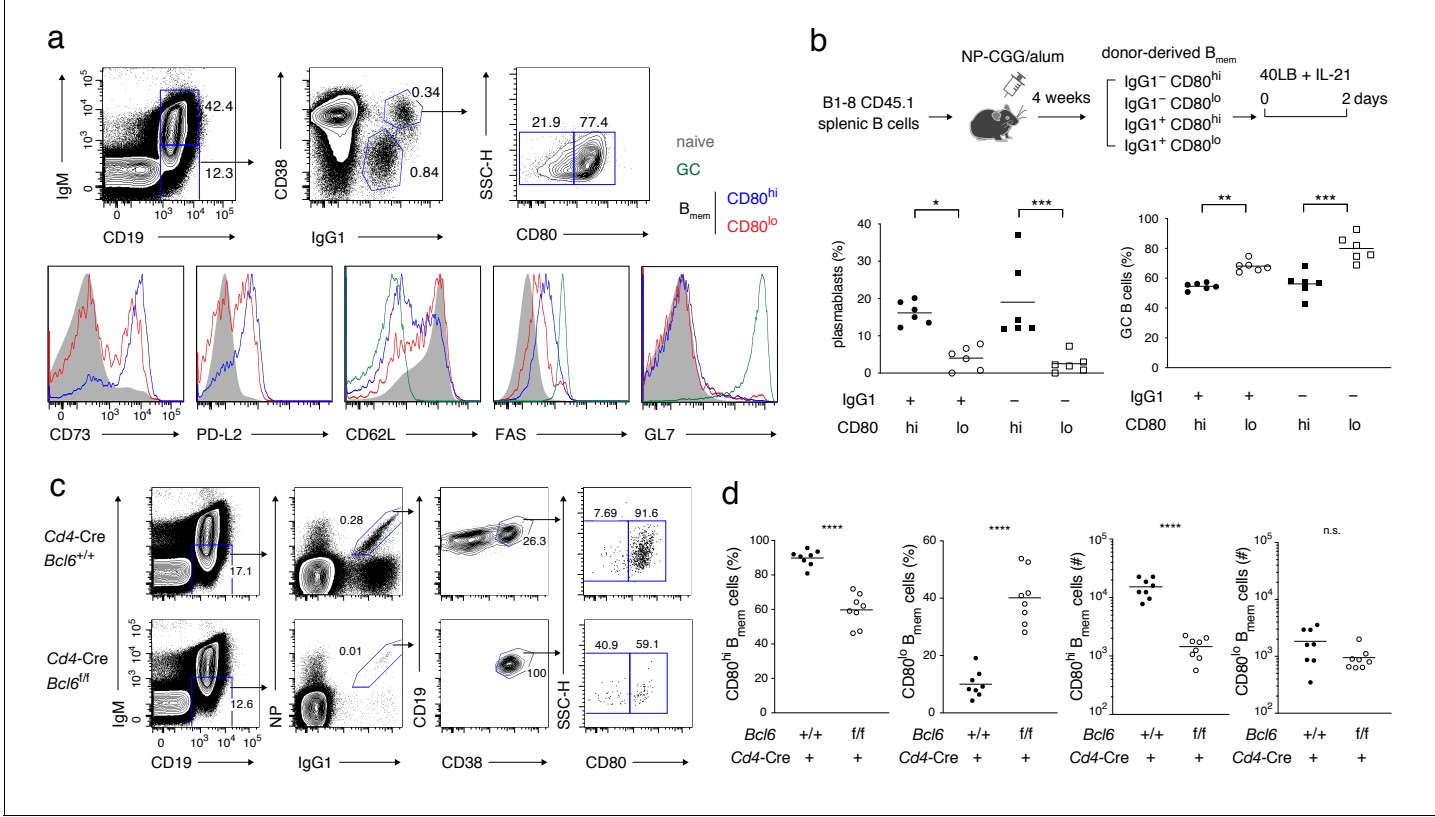

**Figure 1.** Characterization of CD80[hi] and CD80[lo] memory B cells. (**a**) Splenocytes from B6 mice immunized with NP-CGG in alum 4 weeks earlier were analyzed by flow cytometry (FCM). The gating strategy for naïve B cells (CD19[+] IgM[+]), GC B cells (CD19[+] IgM[−] IgG1[+] CD38[−]), and CD80[lo] and CD80[hi] B_mem cells (CD19[+] IgM[−] IgG1[+] CD38[+]) is shown in the top panels, and the expression of the indicated cell-surface proteins in each population is shown in the bottom panels. Data are representative of two independent experiments with similar results. (**b**) Outline of the experimental protocol (top). Splenic B cells from CD45.1 B1-8 ki mice were transferred into B6 mice (CD45.2), which were then immunized with NP-CGG in alum. Four weeks later, four subsets of donor-derived B_mem cells (CD45.1[+] CD19[+] CD38[+]), defined by the expression of IgG1 and CD80, were sorted from recipient spleens, cultured on 40LB feeder layers with IL-21 for 2 days, and analyzed by FCM. The frequency of CD138[+] GL7[−] plasmablasts or PCs and CD138[−] GL7[+] GC B cells in each subset is represented by a dot (bottom; combined data from two triplicate experiments). (**c**) Splenocytes from *Cd4*-Cre, *Bcl6*[+/+] or *Bcl6*[f/f] mice immunized with NP-CGG in alum 6 weeks earlier were analyzed by FCM. The representative data indicate the gating strategy with percentages of the gated population. (**d**) The frequency (%) and absolute number (#) of CD80[hi] and CD80[lo] cells among IgG1[+] B_mem cells in each spleen from individual recipient mice, as analyzed in (**c**) (n = 8). The mean of the values in each group is indicated by a horizontal bar (**b, d**). n.s., not significant (p>0.05); *, p<0.05; ***, p<0.001; ****, p<0.0001; as determined by one-way ANOVA followed by Tukey's multiple comparisons test (**b**) and unpaired Student's *t* test (**d**). All data are representative of two independent experiments, except (**b** and **d**), where data from two independent experiments are combined.

DOI: https://doi.org/10.7554/eLife.44245.002

The following source data and figure supplement are available for figure 1:

**Source data 1.** Source data for *Figure 1b and d*.
DOI: https://doi.org/10.7554/eLife.44245.004
**Figure supplement 1.** Supplementary data for *Figure 1*.
DOI: https://doi.org/10.7554/eLife.44245.003

plasmablasts or PCs and less into GL7[+] GC-like B cells, as compared with CD80[lo] B_mem cells, regardless of their BCR isotype (*Figure 1b* and *Figure 1—figure supplement 1a,b*). These in vitro data were consistent with the previous in vivo data (*Zuccarino-Catania et al., 2014*), and further revealed that the biased differentiation of the CD80[hi] or CD80[lo] B_mem cells is determined in a cell-intrinsic manner, and is essentially independent of BCR isotype and BCR affinity for antigen.

## Strong CD40 signaling induced by $T_{FH}$ cells is required for the development of CD80$^{hi}$ B$_{mem}$ cells

We next sought to clarify a need for GC in the development of CD80$^{hi}$ and CD80$^{lo}$ B$_{mem}$ cells. A previous report indicated that CD80 and PD-L2 were expressed at normal levels on B$_{mem}$ cells in B-cell-specific BCL6-deficient mice that lack GCs (*Kaji et al., 2012*). To examine a role for the GC environment in B$_{mem}$ cell development from normal B cells, we used CD4$^+$ T-cell-specific BCL6-deficient mice, which lack T$_{FH}$ cells and GCs (*Kaji et al., 2012*). Six weeks after immunization, the number of CD80$^{hi}$ B$_{mem}$ cells decreased by approximately ten-fold in *Cd4*-Cre *Bcl6*$^{f/f}$ mice as compared to the control *Cd4*-Cre *Bcl6*$^{+/+}$ mice, while the number of CD80$^{lo}$ B$_{mem}$ cells was essentially unchanged (*Figure 1c,d*). These data suggested that the GC environment, or more specifically T$_{FH}$ cells, facilitate the development of CD80$^{hi}$ B$_{mem}$ cells.

T$_{FH}$ cells differ from naïve or effecter CD4$^+$ T cells in that they express a much higher level of CD40L (*Breitfeld et al., 2000*), as we confirmed using purified T-cell subsets (*Figure 2a* and *Figure 2—figure supplement 1a*). As previously indicated (*Lenschow et al., 1994*; *Ranheim and Kipps, 1993*), stimulation of B cells through CD40, but not BCR, induced CD80 expression in a time- and dose-dependent manner (*Figure 2—figure supplement 1b,c*). Supposing that CD80 expression on B cells that is induced during the primary response is maintained until the B$_{mem}$ cell stage, we hypothesized that T$_{FH}$ cell stimulation through CD40 promotes CD80$^{hi}$ B$_{mem}$ cell development. To test this, we first used CD40L-deficient mice as recipients of antigen-specific (BCR-knock-in) naïve B cells. After immunization of such mice, however, B$_{mem}$ cell development from the donor B cells was abrogated altogether in the absence of CD40L, indicating that CD40L-mediated stimulation is indispensable for B$_{mem}$ cell development (*Figure 2b*).

Next, we treated immunized mice with anti-CD40L blocking antibody (Ab) in a dose that we had determined only partially inhibited Ab production and GC B-cell formation (*Figure 2—figure supplement 1d,e*). This treatment preferentially affected CD80$^{hi}$ B$_{mem}$ cell development. At 10 days after immunization, the frequency of CD80$^{hi}$ B$_{mem}$ cells was significantly reduced among class-switched B$_{mem}$ cells, of which ~90% were IgG1$^+$ and ~10% were IgM$^-$ IgG1$^-$, both normally containing the CD80$^{hi}$ cells at a similar frequency (*Figure 2c,d* and *Figure 2—figure supplement 1f–h*). The absolute number of the CD80$^{hi}$ B$_{mem}$ cells was also severely reduced while there was only a moderate reduction in the absolute number of CD80$^{lo}$ B$_{mem}$ cells, in a condition where the number of GC B cells was reduced by about ten-fold (*Figure 2d* and *Figure 2—figure supplement 1e*). The frequency and the number of the CD80$^{hi}$ B$_{mem}$ cells were somewhat recovered by 6 weeks after immunization, presumably because of generation of such cells in the late GC after the lapse of the injected anti-CD40L Ab (*Figure 2d*). In order to focus on B$_{mem}$ cell generation, avoiding possible effects of alteration in GC formation or B$_{mem}$ cell maintenance, we hereafter mainly analyzed B$_{mem}$ cells by 10 days after immunization, referring to previous reports (*Suan et al., 2017*; *Wang et al., 2017*; *Weisel et al., 2016*). As an opposite experiment, administration of agonistic anti-CD40 Ab to immunized mice markedly increased the frequency and the number of CD80$^{hi}$ B$_{mem}$ cells, while the numbers of CD80$^{lo}$ B$_{mem}$ cells, GC B cells and plasmablasts remained unchanged (*Figure 2e,f*).

Finally, antigen (NP)-specific B cells and carrier [ovalbumin (OVA)]-specific (OT-II) CD4$^+$ T cells, which had been transduced with short hairpin (sh) RNA targeting CD40L (shCd40lg) or unrelated control (shCtrl), were co-transferred into B6 mice, which were then immunized with NP-OVA (*Figure 2g*). Among the generated donor-derived class-switched B$_{mem}$ cells, the frequency of CD80$^{hi}$ B$_{mem}$ cells was significantly lower in mice that had received the CD40L-knockdown T cells as compared to those that had received the control T cells (*Figure 2g–i* and *Figure 2—figure supplement 1i*). These data together suggest that, although CD40L-mediated stimulation is required for development of both B$_{mem}$ cell types, stronger CD40 stimulation by T$_{FH}$ cells selectively facilitates the development of CD80$^{hi}$ B$_{mem}$ cells.

## CD40 signal strength in vitro affects differentiation into CD80$^{hi}$ or CD80$^{lo}$ B$_{mem}$ cells in vivo

The data discussed above strongly suggested that quantitative differences in CD40 signaling in B cells during the primary response determine their developmental fate into either CD80$^{hi}$ or CD80$^{lo}$ B$_{mem}$ cells. However, these in vivo experiments cannot exclude the possibility that some other factors that are affected by the CD40/CD40L manipulation might contribute to the fate decision. In

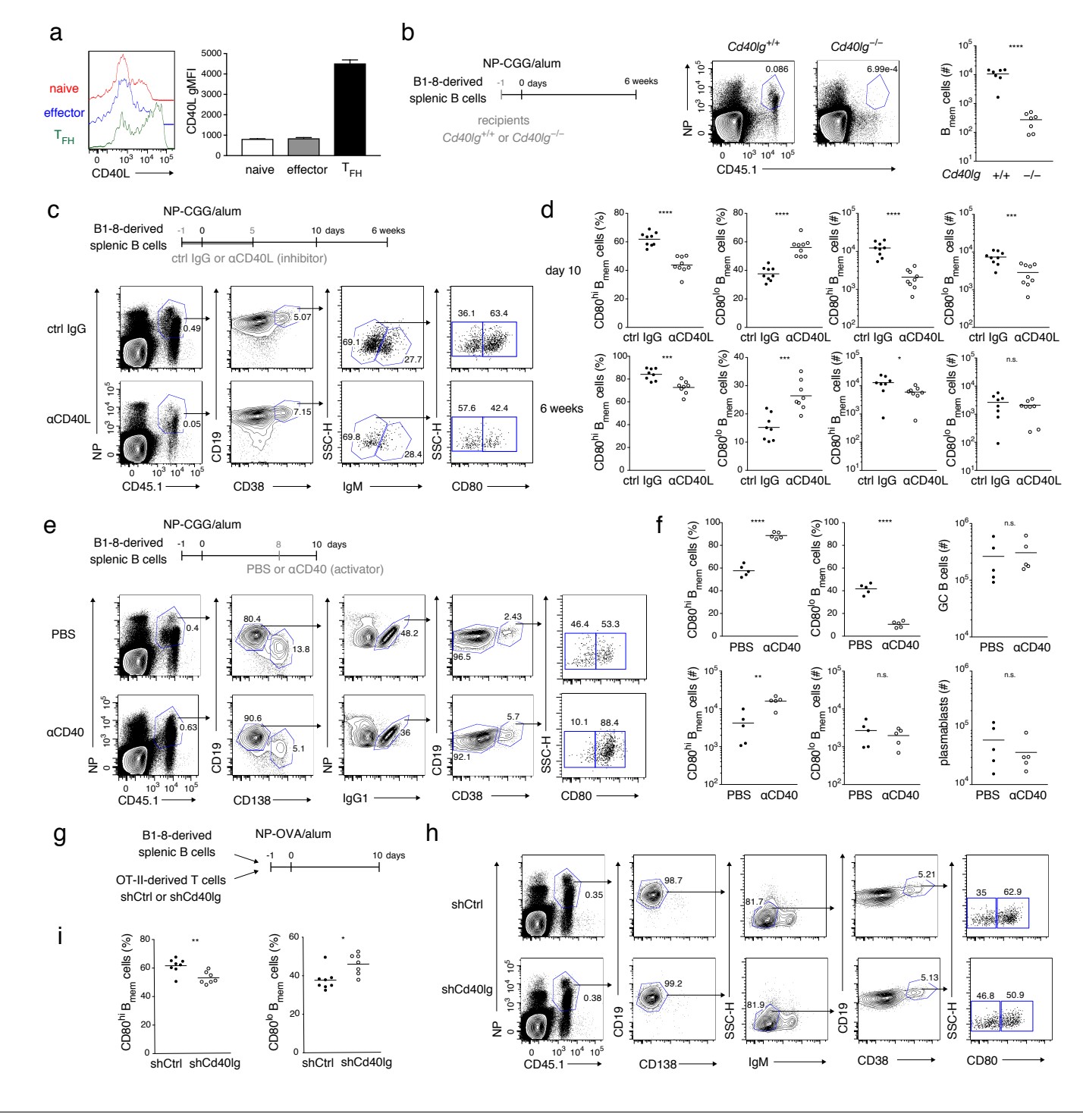

**Figure 2.** Strong CD40 signaling favors CD80$^{hi}$ B$_{mem}$ cell development. (**a**) Naïve T (CD4$^+$ CD62L$^+$ CXCR5$^-$ PD-1$^-$), effector T (CD4$^+$ CD62L$^-$ CXCR5$^-$ PD-1$^-$), and T$_{FH}$ (CD4$^+$ CD62L$^-$ CXCR5$^+$ PD-1$^+$) cells were sorted from spleens of mice immunized with NP-CGG in alum 7 days earlier and then stimulated with phorbol myristate acetate (PMA) and ionomycin for 2 hr. CD40L expression on each cell subset was analyzed by FCM (left) and represented as geometric mean fluorescence intensity (gMFI, right) (mean + s.d. of triplicates). (**b**) *Cd40lg*$^{+/+}$ or *Cd40lg*$^{-/-}$ mice were transferred with splenic B cells from B1-8 ki mice, and immunized with NP-CGG in alum. Six weeks later, the frequency of donor-derived (CD45.1$^+$) NP$^+$ CD19$^+$ B cells (representative data on the left) and the number of the donor-derived B$_{mem}$ cells (CD19$^+$ CD45.1$^+$ NP$^+$ CD38$^+$; plotted on the right; n = 7) in each spleen were analyzed by FCM. (**c, d**) B6 mice were transferred with splenic B cells from B1-8 ki mice, immunized with NP-CGG in alum, and injected subcutaneously (s.c.) with an inhibitory CD40L (MR-1: 1 mg/kg) mAb (αCD40L) or an isotype-matched control (ctrl IgG) Ab every day from day −1 to day

*Figure 2 continued on next page*

*Figure 2 continued*

5 after immunization. Ten days or 6 weeks after immunization, splenocytes of the recipient mice were analyzed by FCM. (c) Representative data (day 10) of the analysis showing the gating strategy. (d) The frequency (%) of CD80$^{hi}$ and CD80$^{lo}$ cells in the donor-derived, class-switched B$_{mem}$ cells (CD45.1$^{+}$ NP$^{+}$ CD19$^{+}$ CD38$^{+}$ IgM$^{-}$), and their absolute numbers (#) at 10 days (top, n = 9) and 6 weeks (bottom, n = 8) after immunization are plotted. (e, f) B6 mice transferred with B1-8 ki B cells and immunized as in (c, d) were injected intraperitoneally (i.p.) with PBS or a stimulatory CD40 mAb (αCD40) (FGK4.5: 250 μg) at 8 days after immunization. Ten days after immunization, splenocytes from the recipient mice were analyzed by FCM. (e) Representative data of the analysis showing the gating strategy. (f) The frequency (%) and absolute numbers (#) of CD80$^{hi}$ and CD80$^{lo}$ cells in the donor-derived, IgG1$^{+}$ B$_{mem}$ cells (CD45.1$^{+}$ NP$^{+}$ CD138$^{-}$ CD19$^{+}$ IgG1$^{+}$ CD38$^{+}$), and the numbers of the donor-derived GC B cells (CD45.1$^{+}$ NP$^{+}$ CD19$^{+}$ IgG1$^{+}$ CD38$^{-}$) or of plasmablasts (CD45.1$^{+}$ NP$^{+}$ CD138$^{+}$) in splenocytes are plotted (n = 5). (g–i) B6 mice, co-transferred on day −1 with B1-8 ki B cells (1 × 10$^{5}$) and OT-II T cells (1 × 10$^{5}$) that had been transduced with control (shCtrl) or shCd40lg retroviral vectors on the previous day, were immunized with NP-OVA in alum. Ten days after immunization, spleen cells from the recipient mice were analyzed by FCM. (g) Outline of the experimental protocol. (h) Representative data showing the gating strategy. (i) The frequencies of CD80$^{hi}$ and CD80$^{lo}$ cells among the donor-derived, class-switched B$_{mem}$ cells, defined as in (d) (n = 8). The mean of the values in each group is indicated by a horizontal bar (b, d, f, i). n.s., not significant (p>0.05); *, p<0.05; **, p<0.01; ***, p<0.001; ****, p<0.0001; unpaired Student's *t* test (b, d, f, i). All data are representative of two independent experiments except (b) and (i), where data from two independent experiments are combined.

DOI: https://doi.org/10.7554/eLife.44245.005

The following source data and figure supplement are available for figure 2:

**Source data 1.** Source data for *Figure 2b, d, f and i*.
DOI: https://doi.org/10.7554/eLife.44245.007
**Figure supplement 1.** Supplementary data for *Figure 2*.
DOI: https://doi.org/10.7554/eLife.44245.006

order to demonstrate a direct contribution of CD40 signaling quantity in B cells, we utilized our in vitro induced GC B (iGB) cell culture system, in which naïve B cells massively proliferate, efficiently switch to IgG1, and differentiate into GC-like B (iGB) cells after being cultured with IL-4 on a feeder layer of 40LB cells (*Nojima et al., 2011*). In addition, these iGB cells differentiate into memory-like B cells [termed induced memory B (iMB) cells] in vivo when transferred into irradiated mice (*Nojima et al., 2011*). To stimulate B cells through CD40 at different levels in this in vitro system, we derived 40LB sublines that express CD40L at low, intermediate and high levels, termed 40LB-lo, 40LB-mid, and 40LB-hi, respectively (*Figure 3a*). As expected, B cells cultured on 40LB-hi (iGB-hi cells) expressed the highest level of CD80, whereas those on 40LB-mid (iGB-mid cells) or 40LB-lo (iGB-lo cells) exhibited intermediate or the lowest levels of CD80 expression, respectively (*Figure 3b*). The iGB-hi, iGB-mid and iGB-lo cells underwent class switching to IgG1 or IgE to similar extents and were mostly CD138$^{-}$ (*Figure 3—figure supplement 1a*). It is of note that the iGB-lo cells expressed Fas at a lower level than iGB-hi and iGB-mid cells, whereas they expressed CD38 and CD62L at higher levels than iGB-hi and iGB-mid cells, perhaps reflecting their relatively lower activation status (*Figure 3—figure supplement 1b*).

These iGB cells were then transferred into irradiated mice and, 2 weeks later, B cells in the spleen were analyzed by FCM (*Figure 3c*). The B cells derived from all of these iGB cells were mostly CD38$^{+}$ (iMB) cells and contained similar percentages of IgG1$^{+}$ cells. However, the iMB cells derived from iGB-hi cells (iMB-hi cells) contained almost exclusively CD80$^{hi}$ cells, whereas iMB cells from iGB-lo cells (iMB-lo cells) were dominated by CD80$^{lo}$ cells, irrespective of their BCR isotypes (IgG1$^{+}$ or IgG1$^{-}$). iMB cells derived from iGB-mid cells exhibited an intermediate phenotype (*Figure 3d,e*). In addition, iMB-hi cells expressed PD-L2, CD73 and FAS at higher levels than iMB-lo cells, whereas both expressed equally low levels of GL7 (*Figure 3f*). Thus, iMB-hi and iMB-lo cells phenotypically resembled CD80$^{hi}$ and CD80$^{lo}$ B$_{mem}$ cells, respectively, that are generated in a physiological immune response (*Figure 1a*).

iGB-hi cells grow far more extensively than iGB-lo cells beyond two days of culture (*Figure 3—figure supplement 1c*). As cell cycling may cause epigenetic and transcriptional changes, it is possible that different cell cycles caused by different CD40 signaling strength affected the bifurcated B$_{mem}$ cell development. To address this issue, iMB cells were generated by transferring iGB-lo and iGB-hi cells on day 2 of culturing, when these cells had expanded to similar levels. As a result, about 80% of iMB cells derived from the day 2 iGB-hi cells were CD80$^{hi}$, whereas about 20% from iGB-lo cells were CD80$^{hi}$. These results are reminiscent of those from iMB cells generated from the day 4 iGB cells, except that the iMB cells from day 2 iGB cells contained fewer IgG1$^{+}$ and fewer PD-L2$^{+}$ cells (*Figure 3—figure supplement 1e,f*). iGB-hi cells, rather than iGB-lo cells, tended to dominate the

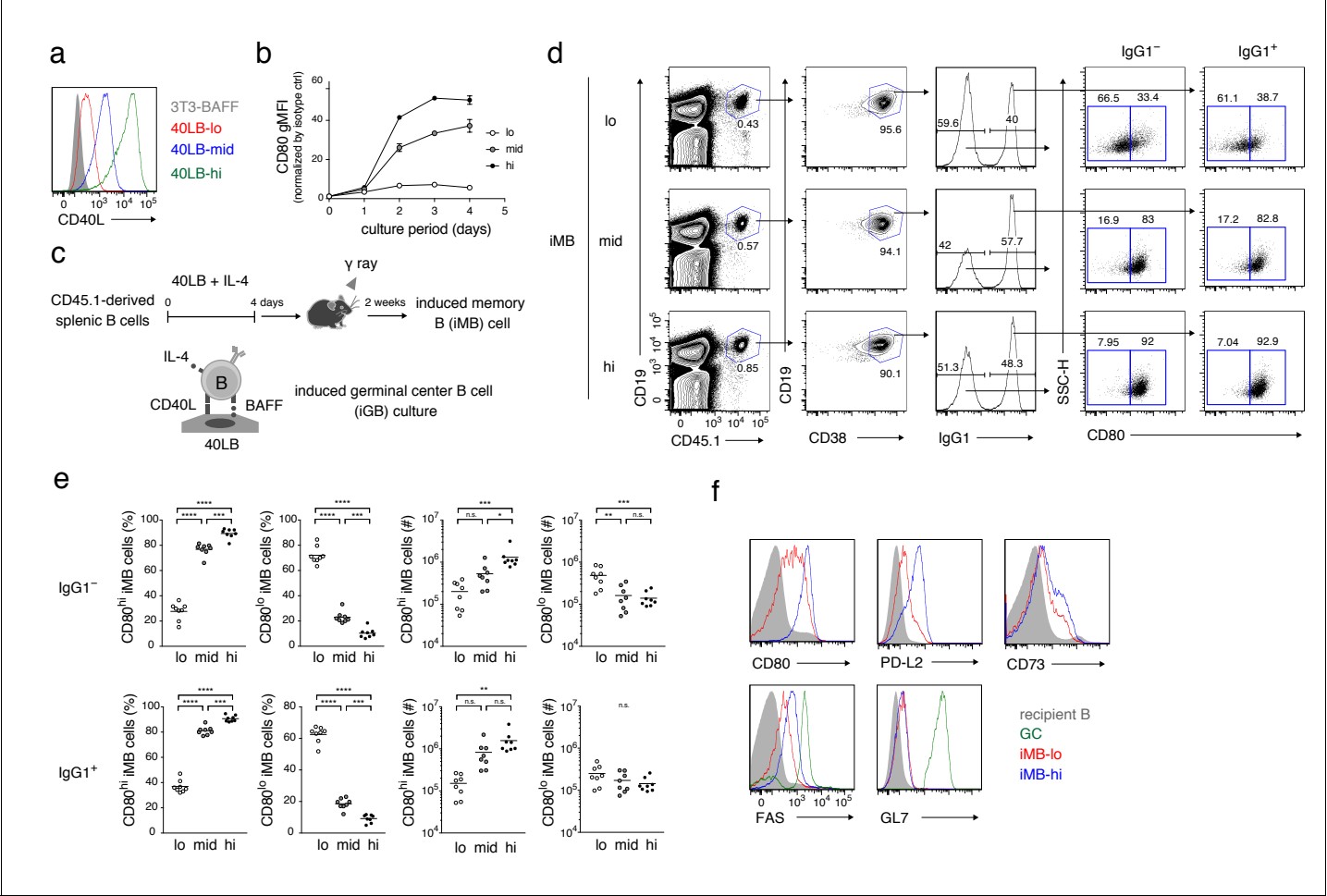

**Figure 3.** The quantity of CD40 signaling determines the differentiation of B cells into distinct B_mem cell subsets. (**a**) Expression of CD40L on 3T3-BAFF cells and 40LB sublines (40LB-lo, 40LB-mid, and 40LB-hi) was analyzed by FCM. (**b**) Splenic B cells were cultured with IL-4 for the indicated number of days on feeder layers of each 40LB subline. The expression of CD80 on the expanded B (iGB) cells was analyzed by FCM and presented as gMFI (mean of triplicates). (**c**) A schematic representation of a method used to generate the induced memory B (iMB) cells. Splenic B cells from CD45.1[+] congenic B6 mice were cultured for 4 days, as in (**b**). The resultant iGB-lo, iGB-mid, or iGB-hi cells were transferred intravenously (i.v.) into γ-irradiated mice (CD45.2[+]), and the donor-derived B_mem-like cells (CD19[+] CD45.1[+] CD38[+]) detected in the recipient spleens 2 weeks after the transfer were designated iMB-lo, iMB-mid or iMB-hi cells, respectively. (**d–f**) Expression of CD80 on IgG1[+] or IgG1[−] iMB cells (iMB-lo, iMB-mid or iMB-hi) generated as in (**c**) was analyzed by FCM. (**d**) Representative data showing the gating strategy. (**e**) The frequencies of CD80[hi] and CD80[lo] cells among the IgG1[+] or IgG1[−] iMB cells (%) and their absolute number per spleen (#) are plotted (n = 8). (**f**) Expression of the indicated surface markers on the recipient total B cells (CD45.1[−] CD19[+]), spontaneous GC B cells (CD45.1[−] CD19[+] CD38[−] GL7[+]), the iMB-lo and the iMB-hi cells (CD19[+] CD45.1[+] CD38[+]). The mean of the values in each group is indicated by a horizontal bar (**e**). n.s., not significant (p>0.05); *, p<0.05; **, p<0.01; ***, p<0.001; ****, p<0.0001; as determined by one-way ANOVA followed by Tukey's multiple comparisons test (**e**). All data are representative of two independent experiments except (**e**), where data from two independent experiments are combined.

DOI: https://doi.org/10.7554/eLife.44245.008

The following source data and figure supplement are available for figure 3:

**Source data 1.** Source data for *Figure 3b and e*.
DOI: https://doi.org/10.7554/eLife.44245.010

**Figure supplement 1.** Cellular phenotype, proliferation capacity, gene expression, and differentiation fate of each type of iGB cells.
DOI: https://doi.org/10.7554/eLife.44245.009

generation of CD80[hi] iMB cells, even when we used day 1 iGB cells (*Figure 3—figure supplement 1g*). These data indicate that the bifurcated B_mem cell fate was not determined by different levels of cell proliferation. However, they also demonstrated that the frequency of the CD80[hi] population in iMB-hi cells increased as the culture period of iGB-hi cells became longer (i.e. from 1 to 2 to 4 days),

thus indicating that the duration of CD40 signaling may also affect the development of CD80$^{hi}$ B$_{mem}$ cells. Although IL-21 is a hallmark T$_{FH}$ cytokine, which is known to support GC B cell proliferation, the addition of IL-21 to the iGB-hi and iGB-lo cell culture did not affect the frequencies of CD80$^{hi}$ iMB cells derived from each type of iGB cells (*Figure 3—figure supplement 1h,i*), suggesting that the contribution of IL-21 to the differential B$_{mem}$ cell fate decision in vivo is less likely.

In order to examine whether the iMB-hi and iMB-lo cells recapitulate the functional differences seen in CD80$^{hi}$ and CD80$^{lo}$ B$_{mem}$ cells, we analyzed their differentiation in culture on the 40LB feeder cells with IL-21. Regardless of their BCR isotype, iMB-lo cells preferentially differentiated toward GC B cells, as clearly seen on day 2, whereas iMB-hi cells preferentially differentiated into plasmablasts or PCs, becoming evident on day 3. The behavior of the iMB-mid cells was intermediate (*Figure 4a,b* and *Figure 4—figure supplement 1*). Next, we investigated the in vivo fate of these iMB cells in response to immunization with a cognate antigen. We sorted allotypically marked NP-binding iMB-lo and iMB-hi cells, derived from iGB-lo and iGB-hi cells of B1-8ki Igκ$^{−/−}$ mice, respectively, and co-transferred the iMB-lo and the iMB-hi cells at an equal ratio into B6 mice together with carrier (CGG)-primed T cells. The recipient mice were immunized with NP-CGG, and their spleen cells were analyzed 4 or 10 days later by FCM (*Figure 4c,d*). Four days after immunization, most donor-derived, NP-binding plasmablasts were derived from iMB-hi cells. By contrast, the vast majority of donor-derived, NP-binding GC B and B$_{mem}$ cells at day 10 were found to originate from iMB-lo cells (*Figure 4e,f*). These in vitro and in vivo data together indicate that iMB-hi and iMB-lo cells functionally represent CD80$^{hi}$ and CD80$^{lo}$ B$_{mem}$ cells, respectively. Taken together, these data indicate that the quantity of CD40 signaling in B cells determines their differentiation fate toward phenotypically and functionally distinct B$_{mem}$ cell subsets.

## Higher BCR affinity for antigen favors the development of CD80$^{hi}$ B$_{mem}$ cells

On the basis of the data described above, it is likely that the CD40 signaling quantity is primarily determined by the expression level of CD40L on cognate T cells. As CD40L on T cells was shown to be induced in an antigen-dose-dependent manner (*Jaiswal and Croft, 1997*), it seemed plausible that the quantity of antigen presented on B cells would determine the expression level of CD40L on cognate T cells, which in turn determines the differentiation fate toward each B$_{mem}$ subset. To confirm that antigen presentation by B cells induces CD40L on cognate T cells in a dose-dependent manner, OT-II-mouse-derived activated T cells were co-cultured with B cells and various concentrations of OVA peptide (*Figure 5a* and *Figure 5—figure supplement 1*). CD40L expression on the T cells was rapidly induced and its levels positively correlated with antigen dose (*Figure 5b*), as was also the case for CD80 expression on B cells on day 2 (*Figure 5c*). This CD80 induction was suppressed by blocking with anti-CD40L mAb, confirming that the CD40L–CD40 interaction leads to CD80 induction on B cells (*Figure 5d*).

During an immune response, B cells expressing a high-affinity BCR would take up more of the cognate antigen, and thus would present a larger amount of antigenic peptide–MHC complex to cognate T cells (*Schwickert et al., 2011*). This would lead to greater CD40L induction than would occur on T cells interacting with B cells with a lower affinity BCR. To investigate the correlation between BCR affinity and B cell differentiation fate, we used B1-8$^{hi}$ ki mice whose λ$^+$ B cells express BCR with ten-fold higher NP affinity than BCR expressed on λ$^+$ B1-8 ki B cells (*Allen et al., 1988*). B cells from B1-8$^{hi}$ ki and B1-8 ki mice, expressing discriminating allotypic markers, were co-transferred into B6 mice, and immunized with NP-CGG in alum. FCM analysis on day 7 after the immunization revealed that IgG1$^+$ B$_{mem}$ cells developed from B1-8$^{hi}$ ki B cells contained a higher frequency of CD80$^{hi}$ cells than those from B1-8 ki B cells (*Figure 5e,f*).

In the next experiment, we stained B$_{mem}$ cells with NP$_{med}$-APC, allophycocyanin (APC) conjugated with NP at a relatively lower valency, which only binds to high affinity anti-NP BCRs (*Nishimura et al., 2011*). Mice were transferred with B1-8 ki B cells, immunized with NP-CGG in alum, and analyzed by FCM 10 days later. Among donor-derived Igλ$^+$ or IgG1$^+$ B$_{mem}$ cells, those stained more brightly with NP$_{med}$ (NP$_{med}$$^{hi}$) contained a higher frequency of CD80$^{hi}$ B$_{mem}$ cells than those stained less brightly (NP$_{med}$$^{lo}$) (*Figure 5g–j*). These data indicated that B cells with higher antigen affinity preferentially differentiate into CD80$^{hi}$ B$_{mem}$ cells rather than CD80$^{lo}$ B$_{mem}$ cells, probably through more extensive antigen presentation to cognate T cells, which results in greater induction of CD40L. Alternatively, B cells with lower affinity may be excluded from the GC and

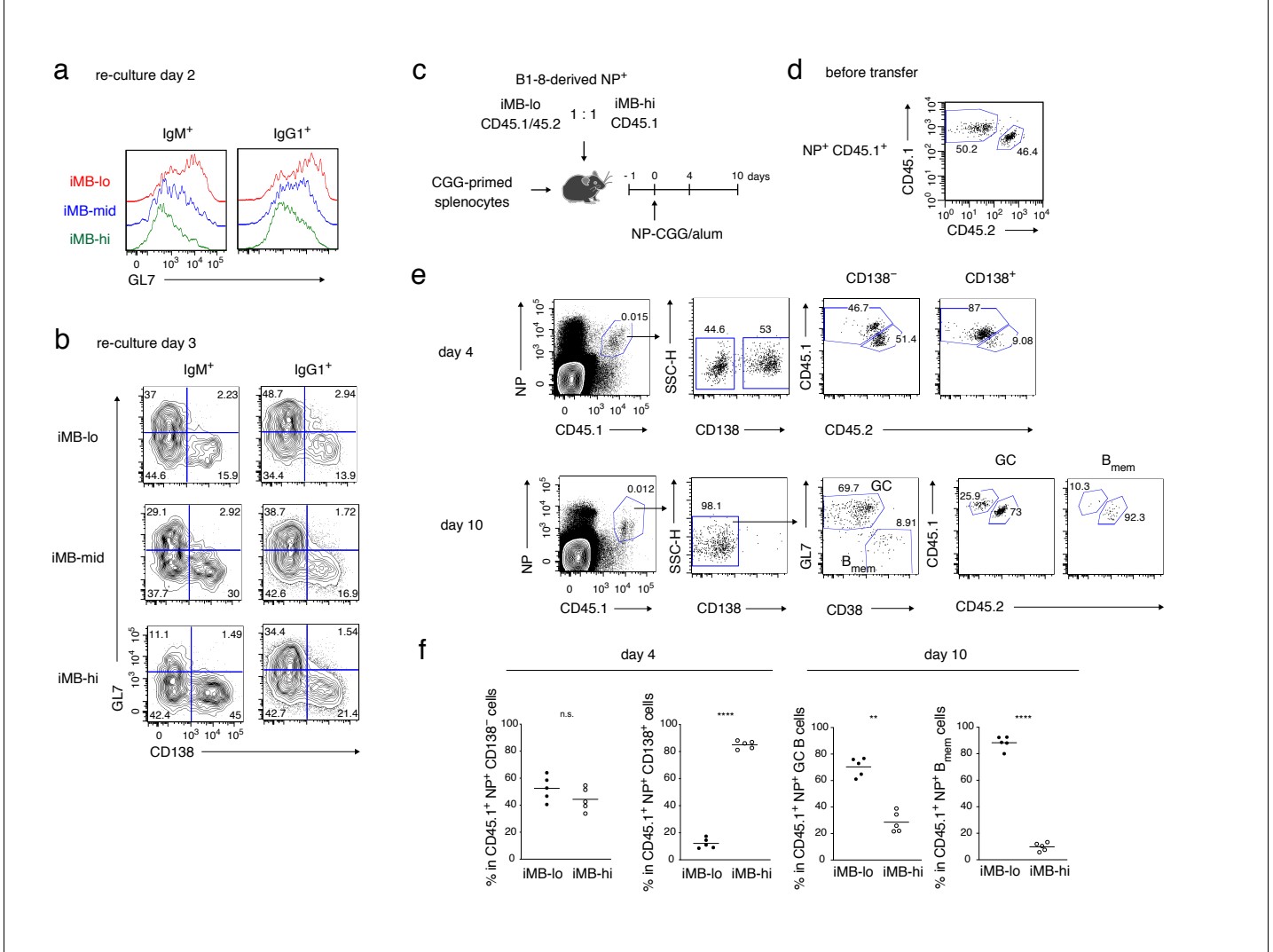

**Figure 4.** Primary CD40 signaling strength affects secondary $B_{mem}$ cell differentiation, either to PCs or to GC B cells. (a, b) Splenic B cells from each recipient mouse (containing iMB cells), generated as in *Figure 3* (d), were cultured on 40LB feeder layers with IL-21 for 2 (a) or 3 (b) days. The expression of GL7 (a, b) and CD138 (b) on gated $IgM^+$ or $IgG1^+$ $CD45.1^+$ (iMB cell-derived) cells was analyzed by FCM. (c–f) iMB-lo and iMB-hi cells were generated from B1-8 ki $Ig\kappa^{-/-}$ CD45.1/CD45.2 or B1-8 ki $Ig\kappa^{-/-}$ CD45.1 iGB cells, respectively, as in *Figure 3* (d). $2.5 \times 10^4$ (for 'day 4') or $1 \times 10^4$ (for 'day 10') of $NP^+$ iMB-lo and iMB-hi cells were mixed and co-transferred into WT B6 recipient mice with $1 \times 10^7$ CGG-primed splenocytes. The recipient mice were then immunized with NP-CGG in alum and analyzed 4 or 10 days after immunization. (c) A schematic of the experimental procedure. (d) A representative FCM profile of the mixture of iMB-lo ($CD45.2^+$) and iMB-hi ($CD45.2^-$) cells, gated on $CD45.1^+$ $NP^+$ cells, before the transfer. (e) Representative FCM data at day 4 and day 10 after immunization showing the gating strategy. (f) The frequencies of iMB-lo- and iMB-hi-derived cells among $CD45.1^+$ $NP^+$ $CD138^-$ or $CD138^+$ cells at day 4, and among $CD45.1^+$ $NP^+$ $B_{mem}$ cells ($CD138^-$ $GL7^-$ $CD38^+$) or GC B cells ($CD138^-$ $GL7^+$ $CD38^-$) at day 10. The mean of the values in each group is indicated by a horizontal bar (f). n.s., not significant (p>0.05); **, p<0.01; ****, p<0.0001; as determined by paired Student's *t* tests (f). All data are representative of two independent experiments.

DOI: https://doi.org/10.7554/eLife.44245.011

The following source data and figure supplement are available for figure 4:

**Source data 1.** Source data for *Figure 4f*.

DOI: https://doi.org/10.7554/eLife.44245.013

**Figure supplement 1.** A gating strategy for IgM+ and IgG1+ cells used in *Figure 4b*.

DOI: https://doi.org/10.7554/eLife.44245.012

therefore fail to access to CD40L on $T_{FH}$ cells. In any case, BCR affinity appears to be a primary determinant for the differential $B_{mem}$ subset development that is dependent on CD40 signaling quantity.

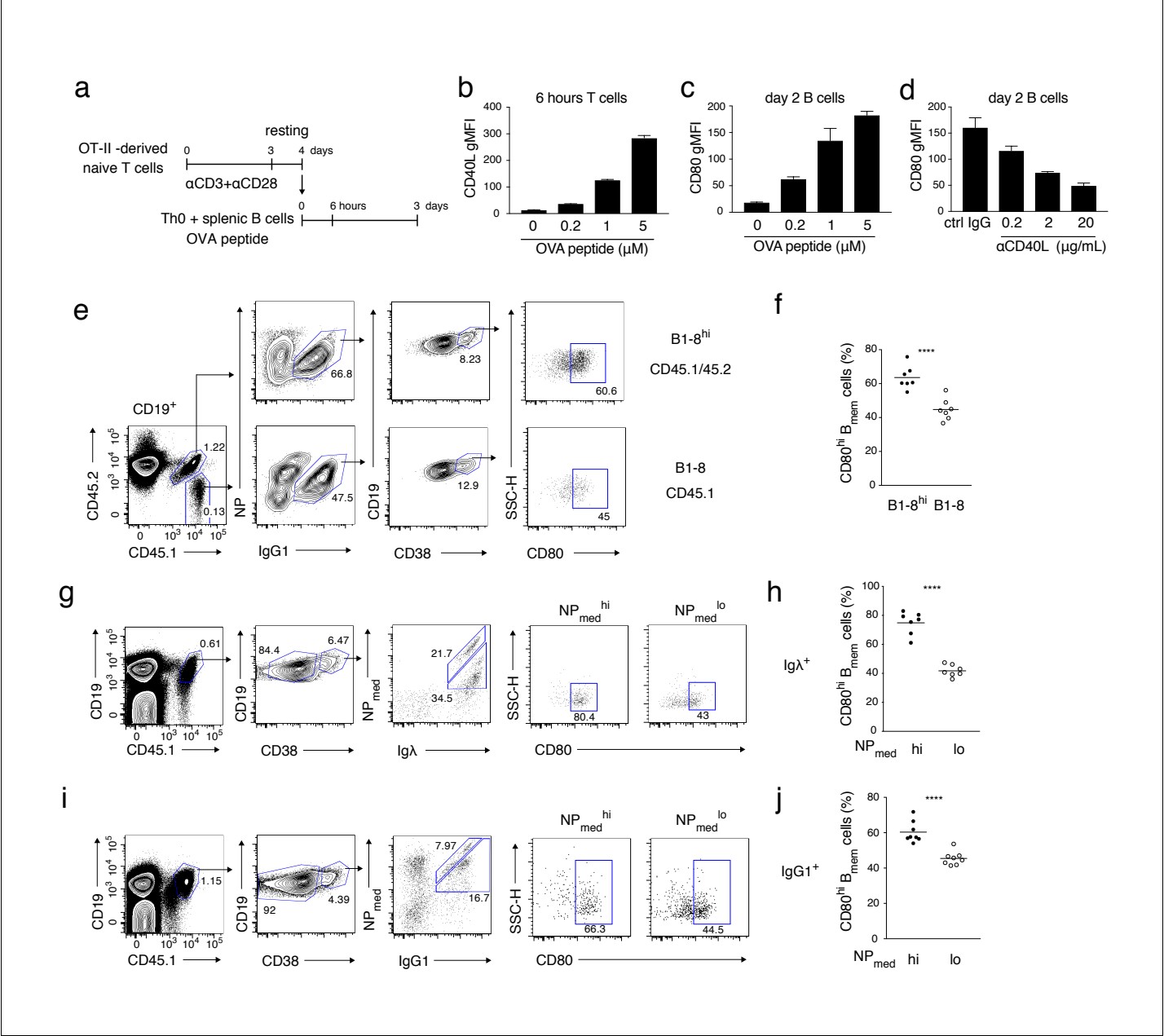

**Figure 5.** High-affinity B cells preferentially differentiate into CD80$^{hi}$ B$_{mem}$ cells, possibly through stronger induction of CD40L on cognate T cells. (a–d) OT-II-derived Th0 cells and splenic B cells were co-cultured with the indicated concentration of OVA peptide for 6 hr or 2 days and analyzed by FCM. (a) An outline of the procedure for the T-B co-culture (see Materials and methods). (b) Expression levels of CD40L on CD4$^+$ T cells after 6 hr co-culture are presented as gMFI (mean + s.d. of triplicates). (c) Expression levels of CD80 on CD19$^+$ B cells after 2 days co-culture are presented as gMFI (mean + s.d. of triplicates). (d) Expression levels of CD80 on B cells after 2 days co-culture with 5 µM OVA peptide in the presence of the indicated concentration of anti-CD40L blocking Ab. Data are shown as in (c). (e, f) 1 × 10$^5$ NP$^+$ splenic B cells from B1-8$^{hi}$ ki (CD45.1/45.2) or B1-8 ki (CD45.1) mice were co-transferred into the recipient B6 mice, which were immunized with NP-CGG in alum on the next day and analyzed by FCM 7 days later. (e) Representative FCM data showing the gating strategy. (f) The frequency (%) of CD80$^{hi}$ among IgG1$^+$ B$_{mem}$ cells (CD19$^+$ CD38$^+$) derived from either B1-8$^{hi}$ ki or B1-8 ki cells is plotted (n = 7). (g–j) B6 mice transferred with B1-8 ki B cells and immunized as in (e, f) were analyzed by FCM at 10 days after immunization. (g, i) Representative FCM data showing the gating strategy. (h, j) The frequencies (%) of CD80$^{hi}$ cells among NP$_{med}$$^{hi}$ or NP$_{med}$$^{lo}$ cells in Igλ$^+$ (h) or IgG1$^+$ (j) B$_{mem}$ cells (CD19$^+$ CD45.1$^+$ CD38$^+$), gating of each as shown in (g) and (i), respectively (n = 8). The mean of the values in each group is indicated by a horizontal bar (f, h, j). ****, p<0.0001; as determined by paired Student's $t$ tests (f, h, j). All data are representative of two independent experiments except (f, h, j), where data from two independent experiments are combined.
DOI: https://doi.org/10.7554/eLife.44245.014

The following source data and figure supplement are available for figure 5:

*Figure 5 continued on next page*

*Figure 5 continued*

**Source data 1.** Source data for *Figure 5b, c, d, f, h, j*.
DOI: https://doi.org/10.7554/eLife.44245.016
**Figure supplement 1.** Representative FCM profile data from the analyses shown in *Figure 5b–d*.
DOI: https://doi.org/10.7554/eLife.44245.015

## Possible mechanisms through which CD40 signaling facilitates GC B-cell differentiation into CD80$^{hi}$ B$_{mem}$ cells

We next investigated the CD40 signaling mechanisms that are responsible for the development of the CD80$^{hi}$ B$_{mem}$ cell subset. NF-κB is a typical transcription factor that are induced by CD40 stimulation (*Berberich et al., 1994*), and the p50/p65 heterodimer was reported to bind to the *Cd80* gene locus and to induce CD80 expression in a B cell line after stimulation (*George et al., 2006*). In accord with these data, among the constitutively active (CA) forms of protein kinases that are known to be activated by CD40 stimulation, CA-IKKβ, an activator of the canonical NF-κB pathway (*Mercurio et al., 1997*), but not CA-Akt or CA-MKK4, upregulated CD80 expression on iGB-lo cells (*Figure 6a*). Stimulation of splenic B cells with a higher concentration (10 μg/ml), but not a lower concentration (1 μg/ml), of anti-CD40 Ab induced the nuclear translocation of the NF-κB subunits c-Rel and RelA (*Figure 6b*). Furthermore, knockdown of *Rel* and *Rela* gene expression in iGB-hi cells resulted in downregulation of surface CD80 expression (*Figure 6c,d* and *Figure 6—figure supplement 1a*). These data indicated that CD80 expression on B cells induced by strong CD40 signaling is mediated by NF-κB, c-Rel and RelA. We next examined whether a similar signaling pathway is involved in the generation of B$_{mem}$ cells. By using the iGB cell system, we showed that c-Rel- and Rela-knockdown iGB-hi cells generated fewer CD80$^{hi}$ iMB cells than mock-treated iGB-hi cells in vivo (*Figure 6c–f* and *Figure 6—figure supplement 1b*). Furthermore, we transferred NP-specific B cells transduced with the *Rela*-knockdown vector or a mock vector into mice, which were then immunized with NP-CGG. The knockdown of *Rela* selectively suppressed the development of CD80$^{hi}$ B$_{mem}$ cells among the B cells that had responded to an immunized antigen (*Figure 6g,h* and *Figure 6—figure supplement 1c*). These data indicated that canonical NF-κB signaling plays a role in CD80$^{hi}$ B$_{mem}$ cell development that is facilitated by stronger CD40 stimulation.

It has been observed that CD40 stimulation induces IRF4 in B cells and that bone marrow-derived dendritic cells from IRF4-deficient mice express a reduced level of CD80 upon LPS stimulation (*Saito et al., 2007*; *Suzuki et al., 2004*). It was also reported that transient or intermediate expression of IRF4 induced GC-related genes through the formation of heterodimers with BATF or PU.1, whereas its sustained or high expression induced PC-related genes through an IRF4 homodimer (*Ochiai et al., 2013*). Thus, we examined whether these transcription factors are involved in CD40 signaling in GC B cells. When ex-vivo GC B cells were cultured with anti-CD40 or anti-BCR Abs, of either high or low doses, or with various cytokines, IRF4 expression was found to be upregulated by a high dose of anti-CD40 or anti-BCR Abs, whereas BATF expression was selectively upregulated by a high dose of anti-CD40 Ab (*Figure 7a,b* and *Figure 7—figure supplement 1a*). Then, we tested whether BATF and IRF4 are involved in the induction of CD80 by using tamoxifen-inducible ER$^{T2}$-BATF or ER$^{T2}$-IRF4 constructs. Induced activation of BATF selectively upregulated CD80 expression in iGB-lo cells, although IRF4 alone did not, and co-activation of BATF and IRF4 slightly enhanced CD80 expression (*Figure 7c* and *Figure 7—figure supplement 1b,c*). In addition, a BATF mutant (BATF-HKE), which is defective in IRF4 binding (*Tussiwand et al., 2012*), failed to upregulate CD80 expression regardless of exogenous IRF4 (*Figure 7d*), suggesting that the exogenous BATF formed a heterodimer with endogenous IRF4 for CD80 upregulation. These data together indicate that the BATF–IRF4 heterodimer that is induced by strong CD40 signaling enhances CD80 expression in activated B cells. As an IKKβ inhibitor suppressed CD40-induced expression of CD80 as well as of BATF and IRF4, the canonical NF-κB pathway appears to upregulate the expression of BATF and IRF4 (*Figure 7e*).

Considering that the NF-κB pathway upregulates CD80 expression on iGB cells and facilitates CD80$^{hi}$ B$_{mem}$ cell development in vivo (*Figure 6*), it is possible that the BATF–IRF4 heterodimer plays a role in the strong CD40 signal that drives GC B cell differentiation into CD80$^{hi}$ B$_{mem}$ cells. This idea was supported by our finding that GL7$^+$ Efnb1$^+$ CD38$^+$ GC-derived memory precursors (pre-

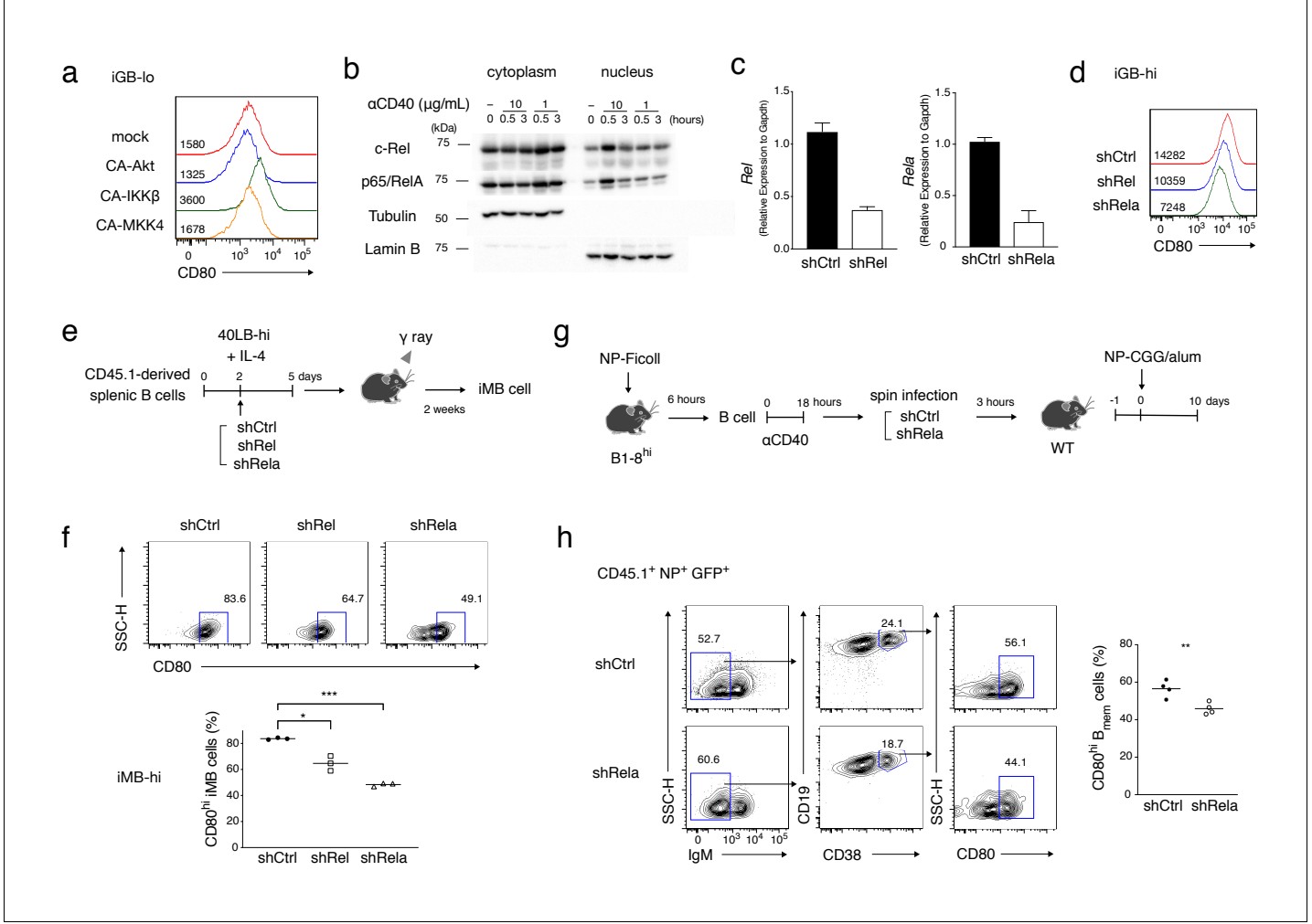

**Figure 6.** NF-κB signaling is involved in CD80hi Bmem cell development. (**a**) Constitutively active (CA) variants of Akt, IKKβ or MKK4 were retrovirally transduced into B cells cultured on 40LB-lo feeder cells (iGB-lo cells) on day 2 of the culture. The expression of CD80 on the gated IgG1+ CD138− infection-marker-positive iGB-lo cells was then analyzed by FCM on day 5. The number on each histogram indicates gMFI. (**b**) Cytoplasmic and nuclear lysates from B cells stimulated with CD40 mAb (1 or 10 mg/ml) for the indicated time periods were analyzed by immunoblotting using Abs against c-Rel and p65/RelA. Tubulin and Lamin B were used as loading controls for cytoplasmic or nuclear proteins, respectively. (**c**) B cells cultured on 40LB-hi feeder cells (iGB-hi cells) were transduced with shCtrl, shRel, or shRela retroviral vectors, each carrying a GFP gene as an infection marker. Three days after the transduction, the expression of *Rel* and *Rela* mRNA in the sorted GFP+ cells was analyzed by qRT-PCR (mean + S.D. of triplicates). (**d**) Expression of CD80 on the GFP+ IgG1+ CD138− iGB-hi cells analyzed by FCM at 3 days after gene transduction, as in (**c**). (**e, f**) The iGB-hi cells transduced with the knock-down constructs as shown in (**c,d**) were transferred into γ-irradiated mice, spleens of which were analyzed by FCM 2 weeks later. (**e**) Outline of the experimental procedure. (**f**) Representative FCM data showing the the expression of CD80 (above) and the frequency (%) of CD80hi cells (bottom; n = 3) in the gene-transduced iMB cells (CD19+ CD45.1+ CD38+ GFP+) formed in the recipients' spleens. (**g, h**) In vivo activated B cells derived from B1-8hi ki mice were transduced with shCtrl or shRela vectors, as described in the Materials and methods, and the resultant B cells (1 × 10^6) were transferred into WT B6 mice. The recipient mice were immunized with NP-CGG in alum on the next day. Splenocytes from these mice were analyzed by FCM at 10 days after immunization. (**g**) Outline of the experimental procedure. (**h**) Representative FCM data showing the gating strategy (left). The frequencies (%) of CD80hi cells among donor-derived, vector-transduced, and class-switched Bmem cells (CD45.1+ NP+ GFP+ IgM− CD19+ CD38+) at 10 days after immunization (right; n = 4). The mean of the values in each group is indicated by a horizontal bar (**f, h**). *, p<0.05; **, p<0.01; ***, p<0.001; as determined by unpaired Student's *t* tests. All data are representative of two independent experiments.

DOI: https://doi.org/10.7554/eLife.44245.017

The following source data and figure supplement are available for figure 6:

**Source data 1.** Source data for *Figure 6c, f and h*.
DOI: https://doi.org/10.7554/eLife.44245.019

**Figure supplement 1.** Gating strategies for the analyses in *Figure 6d and f*, and comlementary data for *Figure 6h*.
DOI: https://doi.org/10.7554/eLife.44245.018

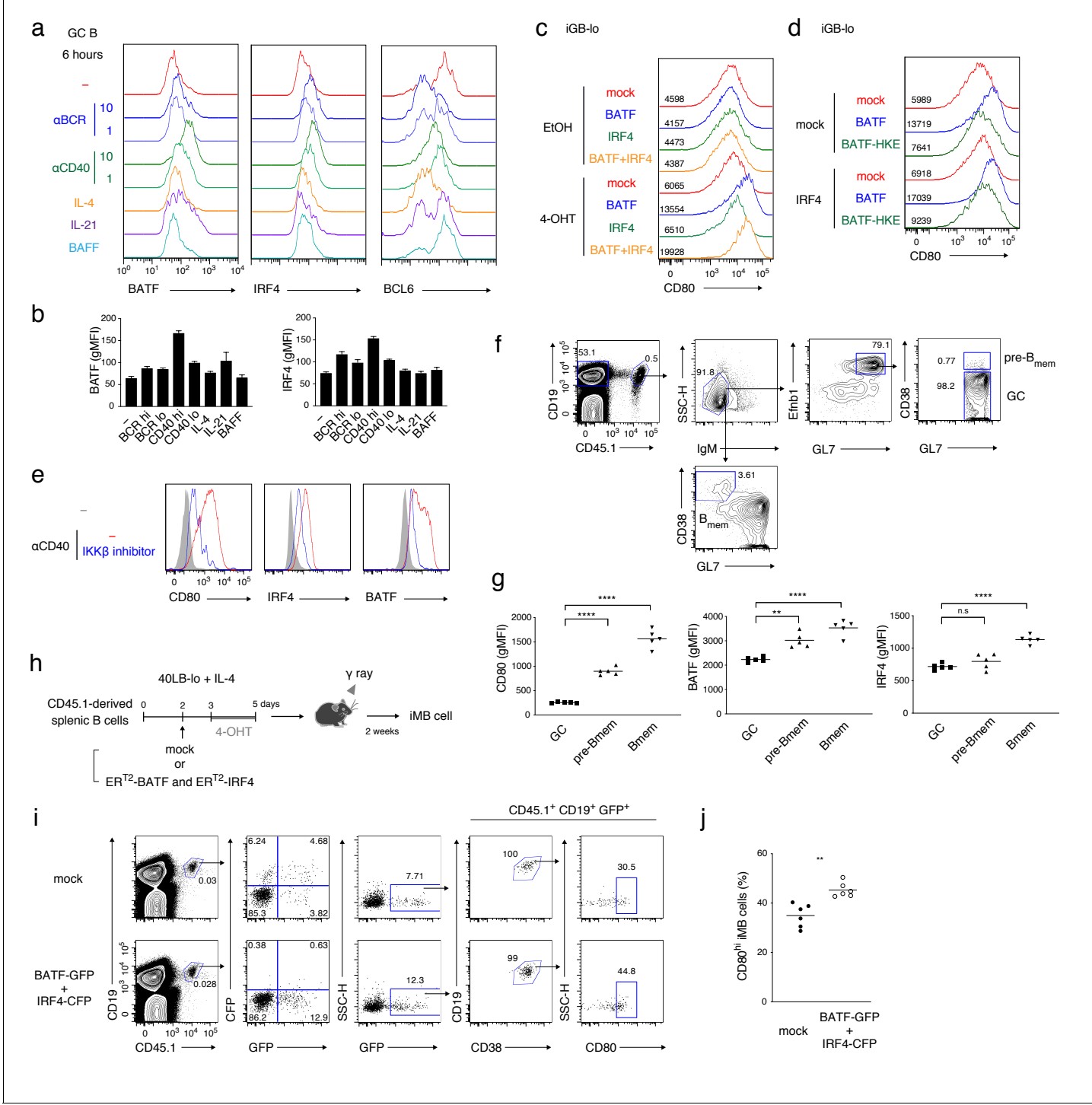

**Figure 7.** CD40-induced BATF may be involved in CD80$^{hi}$ B$_{mem}$ cell development. (**a, b**) GC B cells (CD19$^+$ CD138$^-$ CD38$^-$ GL7$^+$) were sorted from splenocytes of mice at 7 days after immunization with NP-CGG in alum and cultured without (−) or with the addition of the following reagents for 6 hr: anti-IgM plus anti-IgG Abs (αBCR, 10 μg/ml or 1 μg/ml), anti-CD40 Ab (αCD40, 10 μg/ml or 1 μg/ml), IL-4 (10 ng/ml), IL-21 (10 ng/ml) or BAFF (10 ng/ml). Expression of the indicated proteins in these B cells (CD19$^+$ CD138$^-$) was analyzed by intracellular staining followed by FCM. (**a**) Representative FCM data. (**b**) gMFIs of the histograms shown in (**a**). Data are mean + s.d. of triplicates. (**c**) Splenic B cells cultured on 40LB-lo feeder cells (iGB-lo) were transduced with a mock vector or with the indicated vectors expressing each factor fused with ER$^{T2}$ (generated as described in the Materials and methods) on day 2 of the culture, and then treated with vehicle (EtOH) alone or with 4-OHT from day 3 to day 5. The expression of CD80 on these cells was analyzed on day 5, and shown as in (**a**). (**d**) iGB-lo cells were transduced with the indicated combination of the ER$^{T2}$–fusion vectors and treated with 4-OHT as in (**c**). The CD80 expression on these cells is shown as in (**c**). (**e**) Splenic B cells were cultured with anti-CD40 Ab (20 μg/ml)

*Figure 7 continued on next page*

Figure 7 continued
for 2 days without (−) or with an IKKβ inhibitor (BAY11-7082). Expression of the indicated proteins in these cells was analyzed by FCM. The shadowed histograms represent the cells cultured with medium alone. (f, g) Splenic B cells from B1-8 ki CD45.1 mice were transferred into WT B6 mice, which were immunized with NP-CGG in alum on the next day. At 10 days after immunization, splenocytes from the recipients were analyzed by FCM. (f) A representative data showing the gating strategy. (g) gMFI of the indicated proteins in the donor-derived GC B cells (CD19$^+$ CD45.1$^+$ IgM$^-$ GL7$^+$ Ephrin B1$^+$ CD38$^-$), pre-B$_{mem}$ cells (CD19$^+$ CD45.1$^+$ IgM$^-$ GL7$^+$ Ephrin B1$^+$ CD38$^+$), and B$_{mem}$ cells (CD19$^+$ CD45.1$^+$ IgM$^-$ GL7$^-$ CD38$^+$) (n = 5). (h–j) iGB-lo cells were transduced with the retroviral vectors expressing ER$^{T2}$-BATF-ires-GFP and ER$^{T2}$-IRF4-ires-CFP (BATF-GFP +IRF4 CFP), or with empty vectors expressing GFP and CFP (mock) on day 2 of culture treated with 4-OHT from day 3 to day 5, and then transferred into γ-irradiated mice. Two weeks after the transfer, spleen cells of the recipient mice were analyzed by FCM. (h) Outline of the experimental procedure. (i) A representative data showing the gating strategy. (j) The frequency (%) of CD80$^{hi}$ cells in the ER$^{T2}$-BATF gene-transduced iMB cells formed in the recipients' spleens (CD19$^+$ CD45.1$^+$ CD38$^+$ GFP$^+$) (n = 6). The mean of the values in each group is indicated by a horizontal bar (g, j). *, p<0.05; **, p<0.01; as determined paired Student's t tests. All data are representative of two independent experiments except (j), in which data from two independent experiments are combined.
DOI: https://doi.org/10.7554/eLife.44245.020
The following source data and figure supplement are available for figure 7:

Source data 1. Source data for *Figure 7b, g and j*.
DOI: https://doi.org/10.7554/eLife.44245.022
Figure supplement 1. Supplementary data for *Figure 7*.
DOI: https://doi.org/10.7554/eLife.44245.021

B$_{mem}$) cells (*Laidlaw et al., 2017*) expressed BATF and CD80 at higher levels than GL7$^+$ Efnb1$^+$ CD38$^-$ GC B cells at 10 days after immunization (*Figure 7f,g* and *Figure 7—figure supplement 1d*).

Finally, to investigate whether the BATF–IRF4 heterodimer is involved in the development of CD80$^{hi}$ B$_{mem}$ cells, iMB cells were generated from iGB-lo cells transduced with both ER$^{T2}$-BATF and ER$^{T2}$-IRF4, which were then activated in the culture with 4-OHT (*Figure 7h*). In the iMB cells, ER$^{T2}$-IRF4-expressing (CFP$^+$) cells could not be detected, possibly because cells that expressed an excess amount of IRF4 had differentiated into PCs. On the other hand, ER$^{T2}$-BATF-expressing (GFP$^+$) iMB cells were present and contained a significantly higher proportion of CD80$^{hi}$ cells than mock-transduced iMB cells (*Figure 7i,j* and *Figure 7—figure supplement 1e*). Taken together, our data indicate that strong CD40 signaling is converted into the activation of NF-κB and the following upregulation of BATF, resulting in the generation of CD80$^{hi}$ B$_{mem}$ cells.

## Discussion

The regulation of the bidirectional response of B$_{mem}$ cells, either to PCs or to GC B cells upon secondary antigen challenge, has recently been explained by defining the functionally different B$_{mem}$ cell subsets. According to the report by Shlomchik and colleagues, CD80$^+$ PD-L2$^+$ B$_{mem}$ cells preferentially differentiate into PCs, whereas CD80$^-$ PD-L2$^+$ and CD80$^-$ PD-L2$^-$ B$_{mem}$ cells differentiate to GC B cells (*Zuccarino-Catania et al., 2014*). As the class-switched B$_{mem}$ cell population that we mainly focused on is mostly composed of CD80$^+$ PD-L2$^+$ and CD80$^-$ PD-L2$^+$ cells, and also because CD80$^-$ PD-L2$^+$ and CD80$^-$ PD-L2$^-$ B$_{mem}$ cells are functionally similar to each other (*Zuccarino-Catania et al., 2014*), we reasoned that the proposed subsets can simply be distinguished by CD80 expression as CD80$^{hi}$ and CD80$^{lo}$ B$_{mem}$ cells, a distinction that we applied in this study to make a multi-color FCM analyses easier. The CD80$^{hi}$ B$_{mem}$ cells were mostly PD-L2$^+$, CD73$^+$, CD62L$^{lo}$, whereas CD80$^{lo}$ B$_{mem}$ cells included PD-L2$^+$ and PD-L2$^-$ cells, CD73$^+$ and CD73$^-$ cells, and mostly CD62L$^{hi}$, which was largely consistent with previous reports (*Dogan et al., 2009*; *He et al., 2017*; *Pape et al., 2011*; *Zuccarino-Catania et al., 2014*).

Our data showing preferential differentiation of the CD80$^{hi}$ and CD80$^{lo}$ B$_{mem}$ cells into plasmablasts/PCs and GC B cells in vitro, respectively, was also consistent with the previous in vivo data showing the preferential differentiation of each B$_{mem}$ cell subset during the recall response. Reports showing that B$_{mem}$ cells corresponding to the CD80$^{hi}$ B$_{mem}$ cells have BCRs with higher antigen affinity than those corresponding to the CD80$^{lo}$ B$_{mem}$ cells, and that high affinity B cells are more prone to become PCs, seemed to suggest that BCR-signal strength determines the fates of CD80$^{hi}$ and CD80$^{lo}$ B$_{mem}$ cells upon the secondary challenge (*Phan et al., 2006*; *Zuccarino-Catania et al., 2014*). However, our in vitro culture system without specific antigens clearly showed that the distinct fates of these B$_{mem}$ subsets upon re-stimulation are not determined by the BCR

affinity or isotype, although they might be affected by slightly different levels of CD40 expression on CD80$^{hi}$ and CD80$^{lo}$ B$_{mem}$ cells (*Figure 2—figure supplement 1j*). Thus, cell status represented by, for example, transcriptomic or epigenetic profiles may largely define the function of each B$_{mem}$ subset (*He et al., 2017*; *Kometani et al., 2013*; *Zuccarino-Catania et al., 2014*).

Previously it was reported that GC depletion by anti-CD40L mAb treatment reduced the frequency of CD80$^{hi}$ B$_{mem}$ cells, and that the transcriptomic signature of CD80$^{hi}$ B$_{mem}$ cells more closely correlated with CD40-stimulated B cells than did that of CD80$^{lo}$ B$_{mem}$ cells (*He et al., 2017*; *Weisel et al., 2016*). We demonstrated that the development of CD80$^{hi}$ B$_{mem}$ cells was largely dependent on the presence of T$_{FH}$ cells, which express CD40L at a markedly higher level than naïve or effector T cells, and that partial blocking by anti-CD40L Ab or knock-down of CD40L on CD4 T cells during the primary response dominantly affected the generation of CD80$^{hi}$ B$_{mem}$ cells rather than CD80$^{lo}$ B$_{mem}$ cells. Conversely, in vivo stimulation with anti-CD40 Ab during the primary response increased the number of CD80$^{hi}$ B$_{mem}$ cells but not the numbers of CD80$^{lo}$ B$_{mem}$ cells nor GC B cells. As CD40L in the recipient mice was essential for the generation of both types of B$_{mem}$ cells from transferred B cells, we hypothesized that the generation of either CD80$^{hi}$ or CD80$^{lo}$ B$_{mem}$ cells is determined by a difference in the quantity of CD40 signaling.

This hypothesis was strongly supported by our simplified experimental system, which enables in vivo generation of memory-like B (iMB) cells without immunization from naive B cells cultured on feeder cells (40LB) and transferred into mice. Using 40LB cells expressing different levels of CD40L, we clearly showed that stronger in vitro stimulation via CD40 promoted the generation of CD80$^{hi}$ iMB cells, whereas weaker stimulation facilitated the generation of CD80$^{lo}$ iMB cells from B cells with essentially the same BCR repertoire of specificity and isotype. The CD80$^{hi}$ and CD80$^{lo}$ iMB cells phenocopied the CD80$^{hi}$ and CD80$^{lo}$ B$_{mem}$ cells, respectively, in that CD80$^{hi}$ iMB cells preferentially differentiate into plasmablasts/PCs and CD80$^{lo}$ iMB cells into GC B cells (and B$_{mem}$ cells in vivo) in ex-vivo culture and after antigen challenge in vivo. The commitment to differentiate into either CD80$^{hi}$ or CD80$^{lo}$ iMB cells, as determined by the different quantities of CD40 signaling in B cells, was made within two days of culture when the proliferation did not differ among the conditions, and partially made in just one day when CD80 expression was hardly detectable on B cells. Therefore, it seems that the quantity of CD40 signaling in B cells directs cellular programming, which determines the differentiation into distinct B$_{mem}$ cell subsets.

Although the B cells cultured on 40LB feeder cells (iGB cells) mimic some aspect of GC B cells, naturally they differ from genuine GC B cells in that iGB cells are uniformly proliferating, are CD80$^{+}$ (to distinct levels depending on the strength of CD40 stimulation), and do not mutate Ig genes. We consider that the iGB cells that were primarily cultured with IL-4 may represent a certain state of T-cell-activated B cells that are destined to become B$_{mem}$ cells, such as naïve B cells that are activated in the initial phase of primary response or GC B cells that have just undergone selection as what we call pre-B$_{mem}$ cells. Therefore, our data describing the in vivo differentiation of iGB cells into B$_{mem}$-like iMB cells probably explain the mechanism for the induction of bidirectional B$_{mem}$ cell differentiation upon T-B interactions, during both the early phase and the GC phase of the primary immune response. Thus, in either phase, B cells receiving relatively stronger CD40 signaling are committed to CD80$^{hi}$ B$_{mem}$ cells, whereas those receiving weaker CD40 signaling are committed to CD80$^{lo}$ B$_{mem}$ cells. Our additional data suggest that, in the early phase, B cells expressing BCR with higher affinity to antigen present more antigenic peptide to cognate T cells, thus inducing CD40L on these T cells more strongly so that they acquire stronger CD40 signaling than lower affinity B cells. Similarly, in the GC phase, higher-affinity GC B cells dominantly present antigenic peptide, and will acquire more frequent and more durable interactions with T$_{FH}$ cells that express a high level of CD40L (*Figure 8*).

Thus, stronger CD40 signaling in the GC may direct the development of CD80$^{hi}$ B$_{mem}$ cells, as supported by our data. Despite this supposition, it was reported that strong T cell help and CD40 signaling in vivo induce the differentiation of GC B cells into PCs (*Ise et al., 2018*; *Schwickert et al., 2011*), raising a question as to the mechanism for the differentiation of GC B cells into either CD80$^{hi}$ B$_{mem}$ cells or PCs. The fact that CD40 stimulation suppresses PC generation in vitro (*Hawkins et al., 2013*; *Randall et al., 1998*; *Satpathy et al., 2010*) suggests that the strong CD40 signaling alone does not directly promote PC differentiation in the GC. It is possible that additive BCR signaling affects the differentiation of GC B cells into PCs (*Kräutler et al., 2017*), although it has been reported that BCR signaling is inactive in most GC B cells (*Khalil et al., 2012*). Supposing that T$_{FH}$

cells are heterogeneous in terms of cytokine production (*Weinstein et al., 2016*), a cytokine produced from a particular $T_{FH}$ cell subset that interacts with the GC B cells may play a key role. Thus, a combination and integration of signaling pathways, one from the strong CD40 stimulation, the other from a particular cytokine, and maybe more, may ultimately determine the fate of GC B cells. IL-21 is known to induce B cell differentiation into PCs, while IL-21R-deficiency attenuated PC development and accelerated $B_{mem}$ cell development (*Zotos et al., 2010*). In addition, we previously reported that iGB cells that were secondarily cultured with IL-21 preferentially develop into bone marrow PCs but not $B_{mem}$ cells in vivo after adoptive transfer (*Nojima et al., 2011*). Thus, when GC B cells interact with IL-21-producing $T_{FH}$ cells, and receive a strong CD40 signal, they will differentiate into PCs. A $T_{FH}$ cell subset that induces the differentiation of GC B cells into CD80$^{hi}$ $B_{mem}$ cells has yet to be defined. IL-4-producing $T_{FH}$ cells may be this subset, because iGB cells that are cultured with IL-4 on the CD40L$^{high}$ feeder preferentially differentiated into CD80$^{hi}$ iMB cells in vivo. Considering a report showing that $B_{mem}$ cells develop from B cells in the earlier GC, whereas long-lived PCs are generated during the later GC (*Weisel et al., 2016*), it is possible that distinct $T_{FH}$ subsets may work dominantly in B cell selection along the time course of the GC reaction.

Our data demonstrating that CD80$^{lo}$ $B_{mem}$ cell generation was little affected by the absence of GC resulting from $T_{FH}$ cell-deficiency is consistent with a report indicating that the majority of CD80$^{lo}$ PD-L2$^{-}$ $B_{mem}$ cell cells were generated prior to GC formation (*Weisel et al., 2016*). Given that low-affinity B cells do enter into the GC reaction and could maintain their low affinity even after mutation, why are only few CD80$^{lo}$ $B_{mem}$ cells generated during the GC phase? It has been shown that half of GC B cells undergo apoptosis every 6 hr (*Mayer et al., 2017*), and that this response can be avoided by CD40 signaling (*Luo et al., 2018*; *Mayer et al., 2017*). These data suggest that weaker CD40 signaling in the GC phase may not be enough to prevent the apoptosis of B cells and therefore could fail to induce CD80$^{lo}$ $B_{mem}$ cell development. Alternatively, lower-affinity B cells may be excluded from the GC in a competitive situation (*Schwickert et al., 2011*), and therefore fail to receive a strong CD40 stimulation from $T_{FH}$ cells.

As mentioned earlier, CD80$^{hi}$ and CD80$^{lo}$ $B_{mem}$ cells phenotypically and functionally resemble effector memory T ($T_{EM}$) and central memory T ($T_{CM}$) cells, respectively, in that $T_{EM}$ cells are CD62L$^{-}$ and produce abundant effector cytokines upon an antigen re-challenge, whereas $T_{CM}$ cells are CD62L$^{+}$ and have a greater potential for proliferation (*Mueller et al., 2013*). Our finding that CD40 signal strength directs the generation of CD80$^{hi}$ or CD80$^{lo}$ $B_{mem}$ cells also resembles mechanistic aspects of the current model for the generation of $T_{EM}$ and $T_{CM}$ cells: stronger TCR signaling favors $T_{EM}$ cells, whereas weaker TCR signaling favors the generation of $T_{CM}$ cells (*Daniels and Teixeiro, 2015*). It has been reported that the commitment to CD4$^{+}$ $T_{EM}$ or $T_{CM}$ cells is determined by the expression of T-bet and BCL6 transcription factors, respectively (*Pepper et al., 2011*), and that low-affinity TCR signaled greater induction of BCL6 expression but less expression of T-bet compared to high-affinity TCR (*Knudson et al., 2013*). Therefore, it is likely that TCR affinity/signal strength determines the direction of differentiation to distinct $T_{mem}$ subsets, through the induction of distinctive transcription factors (*Daniels and Teixeiro, 2015*).

Previous studies and our observations imply transcriptomic predisposition of CD80$^{hi}$ and CD80$^{lo}$ $B_{mem}$ cells that may account for their preferential differentiation upon re-stimulation, and suggest that the transcriptomic statuses may be established during the primary response as proposed for the T cell memory. It has been demonstrated that the CD40-NF-κB-IRF4 pathway represses the transcription of *Bcl6* (*Saito et al., 2007*), and that *Bcl6* mRNA is more abundantly expressed in CD80$^{-}$ PD-L2$^{-}$ cells than in CD80$^{+}$ PD-L2$^{+}$ cells (*Zuccarino-Catania et al., 2014*). Combined with our data suggesting that NF-κB and the downstream IRF4–BATF heterodimer play a role in generation of CD80$^{hi}$ $B_{mem}$ cells, the development to CD80$^{hi}$ and CD80$^{lo}$ $B_{mem}$ cells is determined by the balance of the expression levels of transcription factors such as IRF4, BATF and BCL6, which are regulated by CD40 signaling quantity.

Canonical NF-κB mainly consists of heterodimer p50/c-Rel or p50/RelA (*Jost and Ruland, 2017*). It has recently been proposed that c-Rel and RelA have different roles in late B cell development: c-Rel promotes proliferation, whereas RelA upregulates Blimp1, a master regulator of PCs (*Heise et al., 2014*; *Roy et al., 2019*). Our data indicate that RelA is contributes more to the generation of CD80$^{hi}$ iMB cells than does c-Rel, which may be involved in determining the nature of CD80$^{hi}$ $B_{mem}$ cells that are predisposed to PC development. Although RelA and c-Rel may function

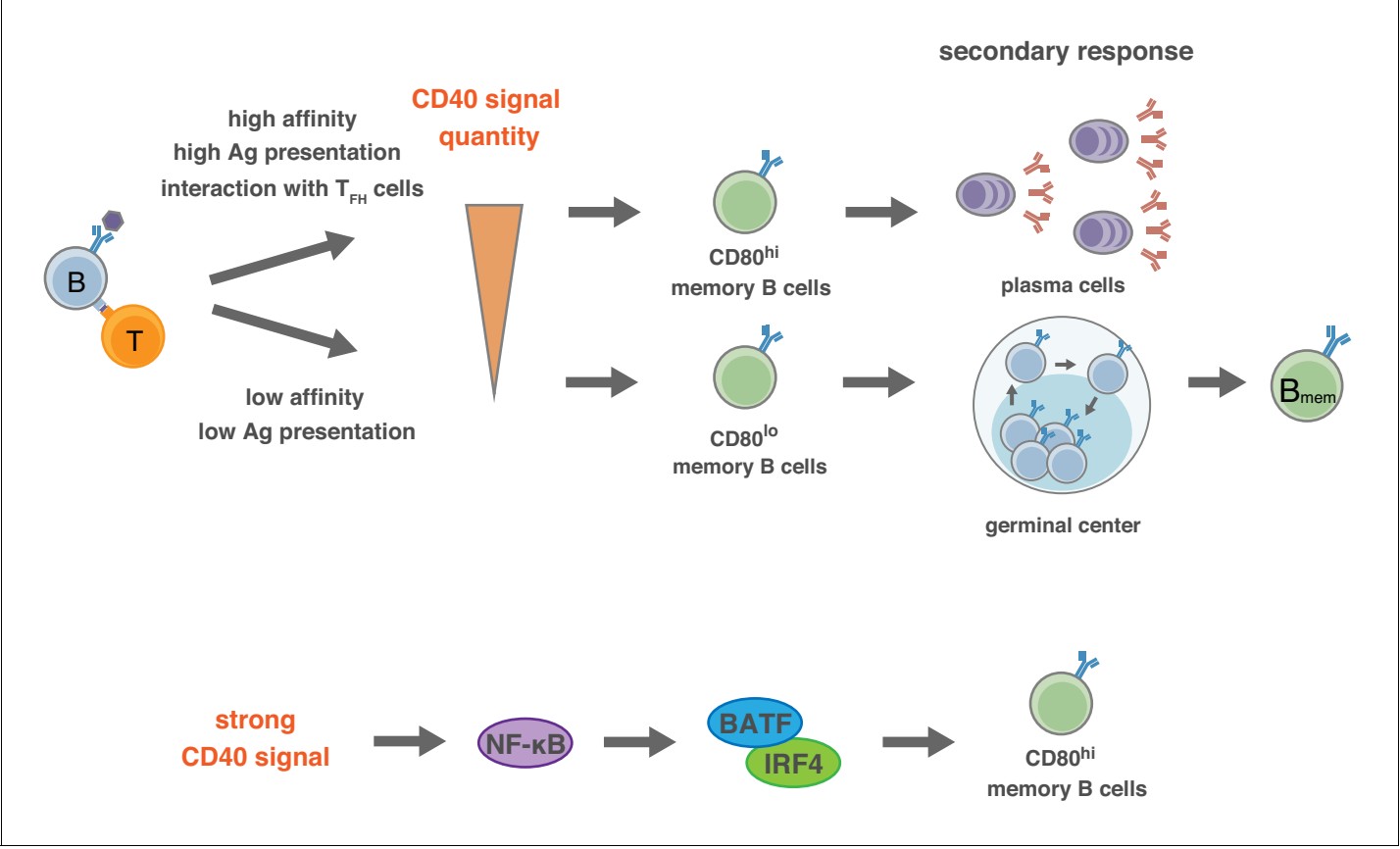

**Figure 8.** Proposed model for the generation of CD80[hi] and CD80[lo] B[mem] cells. (Top) In the pre-GC phase of the primary response, the BCR affinity to antigen or the amount of available antigen determine the quantity of antigen presentation to T cells, and the extent of the induction of CD40L on T cells. Thus, the strength of CD40 signaling in B cells is determined by the interacting T cells, which then directs the differentiation fate to distinct B[mem] subsets: relatively stronger CD40 signal commits B cells towards CD80[hi] B[mem] cells, whereas weaker CD40 signal commits B cells towards CD80[lo] B[mem] cells. After GC formation, T[FH] cells, being able to express a high level of CD40L after TCR stimulation, strongly stimulate CD40 on relatively high-affinity B cells and facilitate their differentiation to CD80[hi] B[mem] cells. (Bottom) Activation of NF-κB, and the downstream BATF–IRF4 heterodimer, may transmit the strong CD40 signaling into a mechanism that facilitates the differentiation towards CD80[hi] B[mem] cells.

DOI: https://doi.org/10.7554/eLife.44245.023

redundantly to some extent to generate CD80[hi] B[mem] cells, our data showing reduced generation of CD80[hi] iMB cells by c-Rel knockdown alone also suggests a unique role for c-Rel.

What would be the survival advantage for individuals of generating bifurcated B[mem] cell subsets upon infection? CD80[hi] B[mem] cells express high-affinity BCR (*Zuccarino-Catania et al., 2014*) and therefore produce high-affinity Abs. On the other hand, CD80[lo] B[mem] cells mainly express low-affinity BCRs, but have a greater proliferative potential and preferentially generate secondary GC upon reencounter with antigen, when they can further diversify their BCR repertoire. Therefore, CD80[lo] B[mem] cells, unlike high-affinity CD80[hi] B[mem] cells, can cope with a broader array of epitopes that may be generated by pathogens through mutations. Taken together, robust and rapid high-affinity Ab production by CD80[hi] B[mem] cells eliminates the majority of reinfecting pathogens, whereas pathogens that escape destruction because of epitope changes resulting from mutations are eliminated by Abs derived from CD80[lo] B[mem] cells that evolved in GCs. Thus, if we could dissect the CD40 signaling pathways that direct differentiation into CD80[hi] B[mem] cells and those promote cell proliferation or survival, selective suppression of the former pathways during vaccination might convert the generation of CD80[hi] B[mem] cells into CD80[lo] B[mem] cells that would eventually produce broadly reactive Abs. Alternatively, additive stimulation of CD40 with proper timing would facilitate the generation of CD80[hi] B[mem] cells that rapidly produce highly specific Abs upon actual infection.

# Materials and methods

**Key resources table**

| Reagent type (species) or resource | Designation | Source or reference | Identifiers | Additional information |
|---|---|---|---|---|
| Genetic reagent (*M. musculus*) | B1-8ki | *Lam et al., 1997* | | Dr. Rajewsky (Max Delbrück Center for Molecular Medicine) |
| Genetic reagent (*M. musculus*) | B1-8$^{hi}$ki | *Shih et al., 2002* | IMSR Cat# JAX:007775; RRID:IMSR_JAX:007775 | Dr. Nussenzweig (The Rockefeller University) |
| Genetic reagent (M. musculus) | *Bcl6*$^{flox}$ | *Kaji et al., 2012* | IMSR Cat# RBRC05663; RRID:IMSR_RBRC05663 | Dr. Takemori (RIKEN) |
| Genetic reagent (*M. musculus*) | *Cd4*-Cre | *Lee et al., 2001* | IMSR Cat# JAX:017336; RRID:IMSR_JAX:017336 | Dr. Kubo (Tokyo University of Science) |
| Genetic reagent (*M. musculus*) | *Cd40lg*$^{-/-}$ | *Xu et al., 1994* | RRID:MGI:2449454 | Dr. Flavell (Yale School of Medicine) |
| Genetic reagent (*M. musculus*) | Igκ$^{-/-}$ | *Chen et al., 1993* | | Dr. Tsubata (Tokyo Medical and Dental University) |
| Genetic reagent (*M. musculus*) | OT-II | *Barnden et al., 1998* | IMSR Cat# JAX:004194; RRID:IMSR_JAX:004194 | Dr. Kubo (Tokyo University of Science) |
| Cell line (*M. musculus*) | 40LB | *Nojima et al., 2011* | | Dr. Kitamura (Tokyo University of Science) |
| Cell line (*M. musculus*) | 40LB-hi | *Takatsuka et al., 2018* | | Dr. Kitamura (Tokyo University of Science) |
| Cell line (*M. musculus*) | 40LB-lo | this paper | | Dr. Kitamura (Tokyo University of Science) |
| Cell line (*M. musculus*) | 40LB-mid | this paper | | Dr. Kitamura (Tokyo University of Science) |
| Antibody | rat monoclonal anti-mouse CD40; FGK4.5 | Bio X Cell | Bio X Cell Cat# BE0016-2; RRID:AB_1107647 | 250 µg |
| Antibody | armenian hamster monoclonal anti-mouse CD40L; MR-1 | *Noelle et al., 1992* | ATCC Cat# HB-11048; RRID:CVCL_8964 | 1 mg/kg; Dr. Abe (Tokyo University of Science) |
| Sequence-based reagent | shRNA | this paper | | See *Supplementary file 1* |
| Software, algorithm | FlowJo | https://www.flowjo.com /solutions/flowjo | RRID:SCR_008520 | |
| Software, algorithm | GraphPad Prism | https://graphpad.com | RRID:SCR_002798 | |

## Mice and immunization

C57BL/6 NCrSlc (B6) mice were purchased from Sankyo Labo Service. All of the following mice were backcrossed to B6 or congenic B6 CD45.1$^+$ mouse strains: B1-8 ki (*Lam et al., 1997*), B1-8$^{hi}$ ki (*Shih et al., 2002*), *Bcl6*$^{f/f}$ (*Kaji et al., 2012*), *Cd4*-Cre (*Lee et al., 2001*), *Cd40lg*$^{-/-}$ (*Xu et al., 1994*), Igκ$^{-/-}$ (*Chen et al., 1993*), and OT-II (*Barnden et al., 1998*). Mice were immunized i.p. with 100 µg of NP$_{32}$-CGG, or NP$_{14}$-OVA where indicated, in alum. Sex-matched, 7-week-old or older mice were used for all experiments. All mice were bred and maintained under specific pathogen-free conditions, and all animal experiments were performed under protocols approved by the

Animal Care and Use Committee of the Tokyo University of Science (Approval No.: S15021, S16019, S17004, S18018).

## Flow cytometry

For all the flow cytometry (FCM) analyses, single-cell suspensions were depleted of red blood cells (RBC) by ammonium chloride lysis, blocked with anti-CD16/32 (FcγRII/III) Ab (2.4G2), and then stained with the appropriate mAbs listed in *Supplementary file 1*, in PBS supplemented with 0.5% BSA, 2 mM EDTA, and 0.05% sodium azide. Stained cells were analyzed using FACSCalibur or FACSCantoII (BD Biosciences) instruments. The data were analyzed using Flowjo (Tree Star). Dead cells, detected by using propidium iodide or Fixable Viability Dye (eBioscience), were gated out in all FCM experiments. For intracellular staining, cells were fixed and permeabilized using a Foxp3 staining kit (eBioscience) before staining.

## Cell purification and culture

Naïve B cells were purified as described previously (*Nojima et al., 2011*). Naïve T cells were purified from OT-II mice as follows: RBC-depleted splenocytes were stained with fluorochrome-conjugated mAbs to CD4, CD25, CD44, and CD62L, and then naïve T cells (CD4$^+$ CD25$^-$ CD44$^-$ CD62L$^+$) were sorted using FACSAriaII or FACSAriaIII (BD Biosciences) instruments. GC B cells were purified from the mice immunized with NP-CGG in alum 7 days previously as follows. Cells from pooled spleens were stained with FITC-conjugated anti-GL7 and anti-FITC microbeads (Miltenyi Biotec), and GL7$^+$ cells were enriched using a MACS system (Miltenyi Biotec). The enriched cells were stained with anti-CD19, anti-CD38 and anti-CD138, and then GC B cells (CD19$^+$ CD38$^-$ CD138$^-$ GL7$^+$) were sorted using FACSAriaII or FACSAriaIII.

B and T cells were cultured in 37°C/5% $CO_2$ conditions in complete medium: RPMI-1640 medium (Wako) supplemented with 10% heat-inactivated fetal bovine serum, 1 mM sodium pyruvate, 50 µM 2-mercaptoethanol, 10 mM HEPES pH7.5, 100 U/ml penicillin and 100 µg/ml streptomycin (GIBCO). Naïve B cells (5 × 10$^6$/ml) were cultured with anti-CD40 (1C10; Southern Biotech) or IKKβ inhibitor (BAY11-7082; Merck). Sorted T cell subsets (2.5 × 10$^5$/ml) were cultured with PMA (20 ng/ml; Sigma) and ionomycin (1 µg/ml; Sigma) for 2 hr. To generate Th0 cells, naïve OT-II T cells (1 × 10$^6$/ml) were cultured in six-well plates (Corning) coated with anti-CD3ε (8 µg/ml; 145–2 C11; Biolegend) and anti-CD28 (8 µg/ml; 37.51; Biolegend) for 3 days, and then cultured without Abs for 1 day. The resultant Th0 cells (1 × 10$^6$/ml) and naïve B cells (1 × 10$^6$/ml) were co-cultured with OVA peptide (*Figure 4a*). GC B cells (1 × 10$^6$/ml) were cultured with anti-IgM (10 or 1 µg/ml; Jackson ImmunoResearch), anti-IgG (10 or 1 µg/ml; Jackson ImmunoResearch), anti-CD40 (1C10; 10 or 1 µg/ml), IL-4 (10 ng/ml; PeproTech), IL-21 (10 ng/ml; PeproTech) or BAFF (10 ng/ml) for 6 hr.

## Adoptive transfer and memory B cell purification

Naïve B cells were purified from B1-8ki CD45.1 mice and the frequency of NP$^+$ cells was determined by FCM. The naïve B cells containing 1 × 10$^4$ NP$^+$ B cells per mouse were transferred into B6 mice, which were then immunized i.p. with NP-CGG in alum on the next day. Four weeks after the immunization, B$_{mem}$ cells were purified from pooled spleens through two-step negative sorting and final positive sorting: cells stained with biotinylated antibodies against CD4, CD8a, CD11b, CD43, CD45.2, CD49b, and Ter119, followed by streptavidin particle DM (BD Biosciences), were negatively sorted sequentially with the iMag (BD) and MACS systems. The resultant cells were stained with fluorochrome-conjugated CD19, CD38, CD45.1 mAbs, and NP-BSA, and then B$_{mem}$ cells (all positive) were sorted using FACSAriaII or FACSAriaII.

## In vivo administration of antibodies

To inhibit CD40 signaling, mice were injected s.c. with the antagonistic CD40L mAb MR-1, (in house; 30 µg per mouse) or with control IgG (IR-AHT-GF, Innovative Research), every day from day −1 to day 5 after immunization. To activate CD40 signaling, mice were injected i.p. with agonistic CD40 mAb (FGK4.5, Bio X Cell; 250 µg per mouse) or PBS at day 8 after immunization.

## Cell lines, iGB cell culture and iMB cell generation

Production of cell lines, iGB cell culture and iMB cell generation were performed as previously described (*Nojima et al., 2011*). 40LB-hi cells were generated by repeated transduction of a CD40L expression vector (pMXs-CD40L-IRES-GFP) into 40LB cells (*Takatsuka et al., 2018*). 40LB-mid and 40LB-lo cells were subclones of 40LB cells made by cell sorting followed by limiting dilution. The CD40L expression level in 40LB-mid cells was equivalent to that of the parental 40LB cells. A parental cell line for 40LB, BALB/c 3T3 fibroblast (clone A31), was provided by RIKEN BRC, Japan. All the BALB/c 3T3-derived cell lines were checked routinely using the PCR Mycoplasma Detection Set (Takara) and proved to be mycoplasma-free.

## Adoptive transfer of iMB cells for immunization

Splenocytes from mice that had been transfected with iGB cells 2 week earlier were analyzed by FCM to estimate the numbers of donor-derived (CD45.1[+]) iMB cells. To examine the response of the iMB cells to a NP antigen in vivo, spleen B cells, including a fixed number of iMB cells derived from B1-8 ki B cells, were co-transferred with CGG-primed spleen cells into WT B6 mice, which were immunized i.p. with NP-CGG in alum on the next day.

## Plasmid constructions

BATF and IRF4 cDNAs were cloned by PCR using iGB cell mRNA. The ER[T2] segment was fused to the 5′-terminus of the BATF or IRF4 cDNAs by ligations using PCR-generated de novo restriction enzyme sites. The BATF-HKE mutant (H55Q, K63D, and E77K) was generated by PCR-based mutagenesis (*Iwata et al., 2017*; *Tussiwand et al., 2012*). Constructs encoding BATF or BATF-HKE, each fused with ER[T2], were cloned into a pMXs-IRES-GFP vector. A construct encoding IRF4 fused with ER[T2] was cloned into a pMXs-IRES-CFP vector, derived from the pMXs-IRES-GFP, in which the GFP sequence was replaced with CFP. CA-IKKβ (S177E and S188E) (*Mercurio et al., 1997*), CA-Akt (E40K) (*Arimura et al., 2004*), and CA-MKK4 (S257E, T261D) constructs were cloned into the pMXs-IRES-GFP. For RNAi, the target sequences of shRNAs, as listed in *Supplementary file 1*, were inserted into a pSIREN-GFP vector, which was made by replacing a puromycin resistance gene in a pSIREN-RetroQ vector (Clontech) with an EGFP sequence.

## Retroviral transduction

Retroviral transduction of iGB cells was performed as previously described (*Haniuda et al., 2016*). For T cells, naïve T cells were stimulated with plate-coated anti-CD3ε (8 μg/mL) and anti-CD28 (8 μg/mL) for 36 hr, and then transduced with retrovirus vectors by spin-infection (*Haniuda et al., 2016*). Retroviral transduction of in-vivo-activated primary B cells and their transfer into mice were performed as previously described (*Inoue et al., 2017*). In brief, B1-8[hi] ki mice were injected i.p. with NP-Ficoll (50 μg), and then B cells were purified from the spleens of these mice 6 hr later and stimulated in vitro with anti-CD40 Ab (2 μg/ml) for 18 hr. Cultured B cells were spin-infected with retroviral vectors and further cultured for 3 hr. The resultant viable B cells ($1 \times 10^6$) were transferred into WT mice for immunization with NP-CGG.

## Immunoblot analysis

Cells were lysed in cytoplasmic extraction (CE) buffer (10 mM HEPES pH 7.9, 10 mM KCl, 0.1 mM EDTA pH 8.0, 0.1 mM EGTA, and 1 mM DTT) for 10 min at 4°C and then NP-40 were added to the final concentration of 0.5%. The cell lysates were centrifuged and supernatants were collected as the cytosolic fraction. Precipitates were washed twice with CE buffer and the final precipitates were lysed in a nuclear extraction buffer (20 mM HEPES pH7.9, 400 mM NaCl, 1 mM EDTA pH 8.0, 1 mM EGTA, 25% glycerol, and 1 mM DTT) for 40 min at 4°C with aggressive mixing every 10 min. The lysates were centrifuged and the supernatants were used as the nuclear fractions. The cytosolic and nuclear fractions were mixed with SDS sample buffer, boiled, and used for SDS-PAGE, followed by immunoblotting using Abs listed in the *Supplementary file 1*.

## Quantitative RT-PCR

The procedures for RNA extraction and reverse transcription to cDNA have been described previously (*Nojima et al., 2011*). Quantitative real-time PCR was performed with a 7500 fast Real-time

PCR system or with QuantStudio 3 (Applied Biosystems). Gene expression levels were determined by the relative standard curve method and normalized to that of *Gapdh*.

## ELISA

NP-specific IgG1 was detected by ELISA using NP-BSA as a plate-coated antigen as described previously (*Nojima et al., 2011*).

## Acknowledgements

We thanks K Rajewsky (Max Delbrück Center for Molecular Medicine) for B1-8ki mice, M Nussenzweig (The Rockefeller University) for B1-8$^{hi}$ki mice, T Takemori (RIKEN) for *Bcl6*-flox mice, R Fravell (Yale School of Medicine) for *Cd40lg*$^{-/-}$ mice, T Tsubata (Tokyo Medical and Dental University) for Igκ$^{-/-}$ mice, T Azuma and Y Tashiro (Research Institute for Biomedical Sciences, RIBS) for NP$_{med}$-APC, J Yagi (Tokyo Women's Medical University School of Medicine) for CA-Akt, M Hibi (Nagoya University) for CA-MKK4, K Haniuda, S Fukao, and S Konishi for plasmid constructs and regents, M Funatsu and M Nomoto for technical assistance, M Kubo, R Goitsuka, and other members of RIBS for technical advice and comments, and P Burrows for critical reading. This works was supported by Grant-in-Aid for Scientific Research (B) (to DK). T Koike is a Research Fellow of Scholarship for Doctoral Students in Immunology of the Japanese Society for Immunology.

## Additional information

### Funding

| Funder | Grant reference number | Author |
| --- | --- | --- |
| Japan Society for the Promotion of Science | 16H05206 | Daisuke Kitamura |

The funders had no role in study design, data collection and interpretation, or the decision to submit the work for publication.

### Author contributions

Takuya Koike, Conceptualization, Data curation, Formal analysis, Validation, Investigation, Methodology, Writing—original draft; Koshi Harada, Investigation; Shu Horiuchi, Supervision, Investigation, Methodology; Daisuke Kitamura, Conceptualization, Supervision, Funding acquisition, Validation, Writing—original draft, Writing—review and editing

### Author ORCIDs

Daisuke Kitamura https://orcid.org/0000-0002-5195-0474

### Ethics

Animal experimentation: All animal experiments were performed under protocols approved by the Animal Care and Use Committee of the Tokyo University of Science (Approval No.: S15021, S16019, S17004, S18018). All surgery was performed under Isoflurane anesthesia, and every effort was made to minimize suffering.

### Decision letter and Author response

Decision letter https://doi.org/10.7554/eLife.44245.027
Author response https://doi.org/10.7554/eLife.44245.028

## Additional files

### Supplementary files

• Supplementary file 1. The list of antibodies and reagents used in this study.
DOI: https://doi.org/10.7554/eLife.44245.024

• Transparent reporting form
DOI: https://doi.org/10.7554/eLife.44245.025

**Data availability**

All data generated or analysed during this study are included in the manuscript and supporting files. Source data files have been provided for Figures 1-7.

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
