## [Decision Letter]

Thank you for submitting your article "The quantity of CD40 signaling determines the differentiation of B cells into functionally distinct memory cell subsets" for consideration by *eLife*. Your article has been reviewed by three peer reviewers, and the evaluation has been overseen by a Reviewing Editor and Tadatsugu Taniguchi as the Senior Editor. The reviewers have opted to remain anonymous.

The reviewers have discussed the reviews with one another and the Reviewing Editor has drafted this decision to help you prepare a revised submission.

Essential revisions:

As you might see from their reports, all three reviewers have found the manuscript interesting and worthy of inviting for a re-submission. Moreover, they have raised several points that if addressed will certainly increase the general impact. In particular, there was a general concern among reviewers about a strict definition of memory B cells as B central memory (B_CM_) and B effector memory (B_EM_) cells. The consensus was that there needs to be an effort to better characterize this population not only in terms of markers (such as surface markers or Bcl6 and Blimp) but also in terms of their time of generation in relation B cell division or after transfer. It will also be beneficial, if within the reach of the authors, to validate their observations in an infectious model and not only with model antigens. In parallel, the text also needs to be softened in several parts. The general view was that the data, as it stands, does not warrant a rigorous definition of B_CM_ and B_EM_ definition as in T cells. In line with this, the manuscript will also benefit from a robust discussion on how the in vitro system does not completely recapitulate the results observed in vivo. A final important point raised by reviewers is how the levels of CD40 stimulation might influence the outcome of B_CM_ and B_EM_. This has to be addressed and discussed extensively. As the manuscript stands, the authors views on this point and the role of CD40 in B cell differentiation appears to contrast with some of the current literature, and this needs to be discussed thoroughly.

Please find below the full reviewers' comments so that you get an idea of their different views as well as experimental work that you might carry out to add to accommodate their concerns.

Reviewer #1:

In this manuscript, Koike et al. investigate the signals involved in the differentiation of distinct populations of memory B cells upon immunization: a CD80-high population that can rapidly differentiate into plasma cells and a CD80-low population that can re-enter germinal centers upon re-challenge. By performing in vivo experiments and taking advantage of an ex vivo germinal center system, authors found that the development of distinct populations of memory B cells is independent of BCR isotypes, but rather associated with the affinity of their BCR and the strength of CD40 signal received from T cells. Authors found that memory B cells with high affinity to antigen (CD80-high), receive strong CD40 signals and preferentially differentiate into effector plasma cells, while low affinity memory B cells (CD80-low) receive less CD40 signal and mostly differentiate into germinal centers. In addition, authors add some mechanistic clues to how CD40 might regulate B cell fate decision through NF-κB. Some of the results present in this study were already shown by the group of Mark Shlomchik (Zuccarino-Catania et al., 2014). However, the present manuscript digs into the signals required for the generation of distinct memory B cell populations. Overall, experiments are well performed, they follow a logic and add sufficiently to our knowledge of memory B cell biology. I think it could be of potential interest for its publication in *eLife* after several changes and additional experiments.

- My main concern relies on the fact that authors try to define these two populations of memory B cells as B central memory (B_CM_) and B effector memory (B_EM_) cells, making a parallel to the memory T cell convention. However, in T cells, these two populations are clearly defined by specific surface markers (for instance, CD44 and CD62L in mice) and are also transcriptionally different. I am not convinced that we are in the presence of two functionally different populations of memory B cells. For instance, even though authors state that B_CM_ are CD80^-^ and B_EM_ are CD80^+^, Figure 1A clearly shows that memory B cells display a gradient of CD80 expression rather than being separated into two distinct populations. In fact, authors arbitrary place a gate and decide which ones will be called CD80^-^ and which ones CD80^+^. If authors wish to create this new nomenclature for memory B cell subsets, they should do a better characterization of these 2 populations, either by flow cytometry or transcriptionally. Authors may combine CD80 with other stainings, like CD62L or PDL2, on the same FACS plot to better separate/define these two populations.

- Authors show that interfering with CD40 signalling affects the balance between CD80-high and CD80-low memory B cells (Figure 2). Do these changes translate into different B cell effector programs (plasma vs germinal center) upon re-challenge?

- All experiments are performed with protein antigens, which raises the question of whether these "two populations" would appear in a real infection model. If that is the case, would they also have distinct CD40 signalling?

Reviewer #2:

The work of Koike et al. starts by reprising work of Shlomchik in describing subsets of memory B cells using CD80 and PDL2. They do add that CD80^hi^ memory B cells are predisposed on restimulation in vitro with CD40L fibroblasts to produce antibody secreting cells (ASC) while CD80^lo^ B_mem_ are predisposed to produce germinal center (GC) cells. They show that GC and T_FH_ are required to bias towards CD80^+^ B_mem_ and that there is a relationship between the amount of CD40L stimulation and proportional representation of the B_mem_ subsets. Diminished amounts of CD40L are associated with reduced CD80^+^ B_mem_. Curiously, when one looks at the different time points when this was assessed, 10 days and 6 weeks, the effect of reduced CD40L becomes less the longer out from the immunisation one goes. This part of the work highlights CD80 as a marker for memory subsets, introduces the potentially confusing (but seductive) nomenclature of central and effector memory B cells and promotes the idea of amount of CD40L exposure as influencing memory subset representation.

The second part is a detailed analysis of the memory B cells that are generated in mice after transfer of B cells activated in vitro with CD40L, IL4, BAFF and fibroblasts, a system called in vitro GC B cells (iGC). This process has been described by this group and involves transferring 20x10^6 activated B cells into an irradiated recipient and waiting two weeks in this case when approximately 10^6 memory B cells are recovered. The authors make the interesting observation that these B_mem_ can be partitioned according to Shlomchik's system and that there is a relationship between the proportion of CD80^hi^ and in vitro exposure to high amounts of CD40L. They also show that these B_mem_ populations show differentiation predispositions that mimic that of the in vivo generated subsets (CD80^+^ to ASC and away from GC; CD80^-^ to GC and away from ASC), although the in vivo test of this differentiation bias (Figure 3I-K) failed to reveal ASC from either subset.

The work then addresses the idea that amount of CD40L a B cell 'sees' in a GC will determine its memory subset outcome by showing, somewhat curiously, that high affinity precursor B cells produce proportionately more CD80^+^ memory B cells at day 7 than low affinity precursor B cells when in competition and that among day 10 memory B cells, high affinity cells are more likely to be in the CD80^+^ subset. This is somewhat complicated by the fact that others using this system (Nussenzweig) have shown that in competition, low affinity cells are excluded from the GC, so one wonders if this is what is being measured here (Figure 4E). Even so, these data are also more or less in agreement with Shlomchik in terms of mutational load and therefore possible affinity improvements. This part of the work also shows that there is a relationship between amount of CD80 upregulation on naïve B cells after their exposure to activated CD4 T cells that is dependent on the amount of peptide the B cells display, providing a link between degree of T cell help and CD80 expression. A major caveat of this conclusion is that this is an in vitro proliferating B cells and not a memory B cell subset.

Finally, the authors look to mechanism and things become a little murky in so far as the confusion between in vitro activated and proliferating B cells and memory subsets remains. The conclusions that NFkB is involved and that its effects are mediated by IRF4 and BATF are to me highly problematic in themselves and also emblematic of my major issue with the work itself. These biochemistry studies, which also ignore a body of work on the consequences of CD40 signalling on GC B cells, are on actively proliferating B cells. One has to seriously question what the relationship is between a B cell cycling in optimal and unchanging conditions in vitro, to that in GC where presumably conditions and stimuli are under constant flux. Indeed, the authors show quite clearly that GC B cells are in fact CD80 negative while all their cells are CD80^+^. It would therefore seem that in vivo, B_mem_ formation is associated with acquisition of CD80 while in the in vitro transfer system, B_mem_ formation is the retention of CD80. Even in this context it is hard to understand how if NFkB is so crucial that knockdown of Rela has such a modest effect (Figure 5F), although the degree of knockdown is not shown. The final figure showing increased expression of CD80 on what are labelled memory precursors, together with increased IRF4 and BATF, is somewhat unconvincing as it is not clear if this amount of CD80 is the baseline for memory B cells, let alone if the slightly increased amounts of IRF4 and BATF are biologically relevant.

My major concern is that this study represents two potentially independent and unrelated observations that are joined by the coincidence of CD80 expression. That is, why should I think that the processes occurring in the in vitro cultures reflect the processes in GC that give rise to memory? In vitro, the B cells are exposed to a constant and unlimited amounts of stimulus, whereas in GC they cycle between exposure to an unknown amount of CD40L, cytokines and cell-cell contact. Equally, IL21 is a crucial component of normal GC function and a defining characteristic of T_FH_, yet it is left out of these cultures. Previously this group reported that the B_mem_ produced by in vivo transfer showed different differentiation potential based on IL21 presence in the first culture. That confuses the CD40L as major factor quite a bit. Similarly, what is the timing of CD80 upregulation in the cultures? Is it immediate and irrelevant to time or is it a timing event? What is the extent of proliferation in the different cultures? Is this phenomenon related to extent of division and would different results be achieved by transferring earlier? Last why are all the mice examined at such early time points? What kind of memory is present at days 7 or 10, and doesn't that change dramatically over the weeks that follow? As shown in this reports Figure 2?

Reviewer #3:

In this manuscript Koike and colleagues investigate the role of CD40-CD40L interaction for the development of functional different memory B cell (MBC) subsets i.e. MBCs that upon re-activation are mostly prone to differentiate into plasma cells (CD80^+^) and those (CD80^-^) mostly prone to re-enter the germinal center (GC) reaction. In light of these functional properties, and in parallel with terms used for T cells, the authors respectively call these subsets effector memory B (B_EM_) and central memory B (B_CM_). Notably, a productive GC reaction is largely required for the formation of the so-called B_EM_, while a GC is largely dispensable for the formation of B_CM_. Overall the study demonstrates that CD40 is required for the formation of MBC cell subsets characterized by CD80^+^ or CD80^-^. The authors also show that B_EM_ formation requires increased CD40 signaling compared to B_CM_ and this correlates to their GC and pre-GC formation, respectively. BCR affinity contributes to the ability of cells to present antigen and therefore receive increased help via CD40, as previously shown by others. The work also shows that once B_EM_ and B_CM_ are formed their functional properties are per se largely independent on BCR receptor signal strength, suggesting a different transcriptomic or epigenetic profile. MBCs displaying IgG1 positivity can also be formed in the absence of a GC reaction, and the authors do not perform experiments where somatic mutation is determined. Does it remain to be shown whether in the GC context the choice of a B cell to differentiate into B_EM_ or B_CM_ exists and if that choice is dependent on CD40 signaling strength. There are additional conceptual and technical aspects of the work that need to be addressed.

1) Figure 1B (Figure 1—figure supplement 1) – The Flow Cytometry gating strategy in Figure 1—figure supplement 1B that pertains the results summarized in Figure 1B needs to be revisited. What is the justification for inclusion of GL7 within the plasma cell gating? Also, GL7 expression within the CD138^-^ subset displays a broad range of intensities and the gating in this case includes GL7 low and likely also GL7 negative cells. How would the results in Figure 1B appear if the gating strategy is more conservative? Staining for Bcl6, as a GC marker, and for Blimp1, as a plasma cell marker, would clarify this issue.

2) Figure 2B – Transfer of B1-8 B cells into CD40lg^-/-^ yields very few cells with an MBC phenotype demonstrating that CD40 engagement is largely required for MBC formation in general. How much or how little do the few formed MBC cells in CD40lg^-/-^ express CD80?

3) Figure 2C, D – What is the impact in GC B cell frequency and numbers and how does that confound the results presented and interpretation? What is the authors interpretation for the cell number recovery of both B_EM_ and B_CM_ by 6 weeks?

4) Figure 3 – How do the varied levels of CD40L expression (40LB-lo, mid, hi) in the 40LB culture system impact on the generation of iGB cells in vitro? What is the phenotype, beyond CD80 expression, of the cells to be transferred into mice? As mentioned previously the study would have increased credibility and interpretations could be made with higher confidence if throughout the study Bcl6, as a GC marker, and for Blimp1, as a plasma cell marker are used.

5) Figure 5E – Also in this experiment it is important to describe the phenotype of the transferred cells. Upon transfer what is the impact of RelA knockdown on B_CM_, the GC reaction, plasma cell formation?

6) Figure 6. What would be the outcome in the formation of either B_EM_ and B_CM_ upon transfer of iGC-lo cells constitutively or conditionally expressing BATF, IRF4 or both?

7) The authors propose, in the fourth paragraph of the Discussion, that the reason why the so-called B_CM_ cells are hardly generated during the GC reaction is because a) these cells fail to induce a high level of CD40L in the cognate T cells and hence would die by apoptosis. b) The authors also write that apoptosis can be avoided by c-Myc expression downstream of CD40 signaling. While the first point (a) is likely to be true given the survival properties of CD40 signaling, the suggestion that c-Myc curtails apoptosis downstream of CD40 (b) is an experimentally unsubstantiated assumption. Current data suggests that c-Myc positivity in the GC reaction represent positive selection downstream of BCR and CD40 signaling and that is most likely the reason why these cells display reduced apoptosis compared to c-Myc negative LZ B cells.

8) The authors go through great lengths to discredit a fundamental role for CD40 in plasma cell differentiation, suggesting that CD40 signaling merely enhances the effect of cytokine from particular T_FH_ cells, likely IL21 expression ones. Clearly this is a gross misinterpretation, previous work has demonstrated that ablation of RelA is critical for GC derived plasma cell formation. Having said this, CD40 signaling may not be per se sufficient; as the authors themselves write IL21 has also been shown to be required for plasma cell formation to ensue. It is not contradictory that CD40 is involved in both MBC and plasma cell formation given that the cell fate is likely to be ultimately determined by a combination and integration of signaling pathways.

9) The authors propose that modulation of CD40 signaling could facilitate the generation of B_CM_ in vaccination possibly beneficial against evolved viruses with epitope mutations. This proposal is however negated by the authors data in Figure 2: if CD40 signaling is impaired (Figure 2C) the formation of both B_EM_ or B_CM_ is impaired (cell numbers); if CD40 signaling is enhanced (Figure 2E) there is no impact on the production of B_CM_ (cell numbers), but a boost occurs in the production of B_EM_ (cell numbers). An avenue with respect to skewing the B_EM_ GC fate to a B_CM_ one may be possible, however as mentioned above, the authors do not perform experiments addressing whether in the GC context a B cell fate choice to B_EM_ or B_CM_ exists and if that choice is dependent on CD40 signaling strength.

---

## [Author Response]

Reviewer #1:In this manuscript, Koike et al. investigate the signals involved in the differentiation of distinct populations of memory B cells upon immunization: a CD80-high population that can rapidly differentiate into plasma cells and a CD80-low population that can re-enter germinal centers upon re-challenge. By performing in vivo experiments and taking advantage of an ex vivo germinal center system, authors found that the development of distinct populations of memory B cells is independent of BCR isotypes, but rather associated with the affinity of their BCR and the strength of CD40 signal received from T cells. Authors found that memory B cells with high affinity to antigen (CD80-high), receive strong CD40 signals and preferentially differentiate into effector plasma cells, while low affinity memory B cells (CD80-low) receive less CD40 signal and mostly differentiate into germinal centers. In addition, authors add some mechanistic clues to how CD40 might regulate B cell fate decision through NF-κB. Some of the results present in this study were already shown by the group of Mark Shlomchik (Zuccarino-Catania et al., 2014). However, the present manuscript digs into the signals required for the generation of distinct memory B cell populations. Overall, experiments are well performed, they follow a logic and add sufficiently to our knowledge of memory B cell biology. I think it could be of potential interest for its publication in eLife after several changes and additional experiments.- My main concern relies on the fact that authors try to define these two populations of memory B cells as B central memory (B_CM_) and B effector memory (BEM) cells, making a parallel to the memory T cell convention. However, in T cells, these two populations are clearly defined by specific surface markers (for instance, CD44 and CD62L in mice) and are also transcriptionally different. I am not convinced that we are in the presence of two functionally different populations of memory B cells. For instance, even though authors state that B_CM_ are CD80- and B_EM_ are CD80^+^, Figure 1A clearly shows that memory B cells display a gradient of CD80 expression rather than being separated into two distinct populations. In fact, authors arbitrary place a gate and decide which ones will be called CD80- and which ones CD80^+^. If authors wish to create this new nomenclature for memory B cell subsets, they should do a better characterization of these 2 populations, either by flow cytometry or transcriptionally. Authors may combine CD80 with other stainings, like CD62L or PDL2, on the same FACS plot to better separate/define these two populations.

In our original manuscript, we intended to simplify the B_mem_ cell subsets defined and extensively characterized by Shlomchik and colleagues (Zuccarino-Catania et al., 2014), namely, CD80^+^ PD-L2^+^, CD80^–^ PD-L2^+^, and CD80^–^ PD-L2^–^, and divided into two (CD80^+^ and CD80^–^), since they demonstrated that class-switched B_mem_ cells were mostly PD-L2^+^, and that the CD80^–^ PD-L2^+^ and CD80^–^ PD-L2^–^ B_mem_ cells were functionally similar. We did not intend to define a new B_mem_ cell subsets. Therefore, agreeing to the reviewer’s comment, we have got rid of the nomenclature, B_EM_ and B_CM_. As the expression of CD80 on B_mem_ cells was not so discrete, as the reviewer pointed out, we decided to call them as CD80^hi^ and CD80^lo^ B_mem_ cells throughout our revised manuscript. We have described above explanation in the first paragraph of the Discussion section of our revised manuscript.

- Authors show that interfering with CD40 signalling affects the balance between CD80-high and CD80-low memory B cells (Figure 2). Do these changes translate into different B cell effector programs (plasma vs germinal center) upon re-challenge?

In the experiment shown in Figure 2C, D, the number of B_mem_ cells in the mice injected with anti-CD40L antibody was too small to perform the secondary transfer for the recall response experiment or ex vivo re-stimulation experiment (as in Figure 1B). Even if the primed mice were directly re-challenged, the markedly lower number of total B_mem_ cells in the antibody-injected mice as compared to the control mice (Figure 2C) would make it difficult to properly compare the numbers of PCs or GC B cells after the secondary challenge. In any case, the in vivo CD40-blocking would inhibit proliferation and therefore affinity maturation of B cells in the GC, and thus reduce the affinity of B_mem_ cells to an antigen as compared to the control, which would affect the outcome of the in vivo recall response. Another experiment using the memory-like iMB^hi^ and iMB^lo^ cells with uniform, unmutated BCRs to examine their recall (-like) response in the same individuals (the former Figure 3I-K; the new Figure 4C-F) clearly demonstrated that the primary CD40 signaling quantity determines the different effector programs of B_mem_ cells that are independent of BCR affinity.

- All experiments are performed with protein antigens, which raises the question of whether these "two populations" would appear in a real infection model. If that is the case, would they also have distinct CD40 signalling?

We agree that the infection model experiments would be informative, but we could not perform such experiments since it will take several months to have an application for such animal experiments approved by our committee and then to perform such experiments.

Reviewer #2:The work of Koike et al. starts by reprising work of Shlomchik in describing subsets of memory B cells using CD80 and PDL2. They do add that CD80^hi^ memory B cells are predisposed on restimulation in vitro with CD40L fibroblasts to produce antibody secreting cells (ASC) while CD80^lo^ B_mem_ are predisposed to produce germinal center (GC) cells. They show that GC and T_FH_ are required to bias towards CD80^+^ B_mem_ and that there is a relationship between the amount of CD40L stimulation and proportional representation of the B_mem_ subsets. Diminished amounts of CD40L are associated with reduced CD80^+^ B_mem_. Curiously, when one looks at the different time points when this was assessed, 10 days and 6 weeks, the effect of reduced CD40L becomes less the longer out from the immunisation one goes. This part of the work highlights CD80 as a marker for memory subsets, introduces the potentially confusing (but seductive) nomenclature of central and effector memory B cells and promotes the idea of amount of CD40L exposure as influencing memory subset representation.

Before responding to each comment, we note here that we have got rid of the nomenclature, B_EM_ and B_CM_, and called them as CD80^hi^ and CD80^lo^ B_mem_ cells throughout our revised manuscript, as mentioned in the response to the first reviewer.

In response to the comment *“Curiously, when one looks at the different time points when this was assessed, 10 days and 6 weeks, the effect of reduced CD40L becomes less the longer out from the immunisation one goes.”*:

As the reviewer commented, the effect of injection of anti-CD40L antibody on the B_mem_ cell development was less impressive at 6 weeks after immunization (Figure 2D). We think this is because CD80^hi^ B_mem_ cells may have been generated in later GC after the injected antibody had lapsed, as we have described in the Results section of the revised manuscript (subsection “Strong CD40 signaling induced by T_FH_ cells is required for development of CD80^hi^ B_mem_ cells”, third paragraph).

The second part is a detailed analysis of the memory B cells that are generated in mice after transfer of B cells activated in vitro with CD40L, IL4, BAFF and fibroblasts, a system called in vitro GC B cells (iGC). This process has been described by this group and involves transferring 20x10^6 activated B cells into an irradiated recipient and waiting two weeks in this case when approximately 10^6 memory B cells are recovered. The authors make the interesting observation that these B_mem_ can be partitioned according to Shlomchik's system and that there is a relationship between the proportion of CD80^hi^ and in vitro exposure to high amounts of CD40L. They also show that these B_mem_ populations show differentiation predispositions that mimic that of the in vivo generated subsets (CD80^+^ to ASC and away from GC; CD80- to GC and away from ASC), although the in vivo test of this differentiation bias (Figure 3I-K) failed to reveal ASC from either subset.

In the experiment shown in the former Figure 3I-K, NP-specific plasmablasts could hardly be detected on day 10 or later in the spleen, but were readily detectable on day 4 after immunization. Thus, we have included the data of the plasmablast analysis on day 4, which shows that vast majority of the donor-derived plasmablasts was derived of iMB-hi cells, as shown in the new Figure 4E, F. We have described about this data in the Results section of the revised manuscript (subsection “CD40 signal strength in vitro affects differentiation into CD80^hi^ or CD80^lo^ B_mem_ cells in vivo”, last paragraph).

The work then addresses the idea that amount of CD40L a B cell 'sees' in a GC will determine its memory subset outcome by showing, somewhat curiously, that high affinity precursor B cells produce proportionately more CD80^+^ memory B cells at day 7 than low affinity precursor B cells when in competition and that among day 10 memory B cells, high affinity cells are more likely to be in the CD80^+^ subset. This is somewhat complicated by the fact that others using this system (Nussenzweig) have shown that in competition, low affinity cells are excluded from the GC, so one wonders if this is what is being measured here (Figure 4E). Even so, these data are also more or less in agreement with Shlomchik in terms of mutational load and therefore possible affinity improvements.

We agree to the possibility that low affinity cells are excluded from the GC and therefore fail to access to CD40L on T_FH_ cells. This possibility does not contradict to our original idea that higher affinity B cells preferentially differentiate into CD80^hi^ B_mem_ cells through getting stronger stimulation with CD40L on T_FH_ cells. Thus, we have included the new possibility and described in the Results section (subsection “Higher BCR affinity for antigen favors development of CD80^hi^ B_mem_ cells”, last paragraph) and the Discussion section (seventh paragraph) of the revised manuscript.

This part of the work also shows that there is a relationship between amount of CD80 upregulation on naïve B cells after their exposure to activated CD4 T cells that is dependent on the amount of peptide the B cells display, providing a link between degree of T cell help and CD80 expression. A major caveat of this conclusion is that this is an in vitro proliferating B cells and not a memory B cell subset.

With the data in the former Figure 4A-D, we intended to show that CD40L expression was induced on activated T cells to the levels in proportion to the extent of antigen presentation on B cells, and that the induced levels of CD40L quantitatively correlated to its function to stimulate cognate B cells through CD40, as assessed by CD80 induction. These data indicate quantitative relationship among B-cell antigen presentation, CD40L expression on T cells and B cell activation, and would serve as an introduction for the following experiments. We did not mean that the CD80 expression levels on naïve B cells directly correlated to the development of B_mem_ cells with distinct expression levels of CD80. However, it is possible that the levels of CD80 expression induced by the T-B interaction during the early primary response or during GC reaction may be maintained through the development to B_mem_ cells. We have included a sentence mentioning this possibility for the logical sequence of the experiments shown in Figure 2, in the Results section of the revised manuscript (subsection “Strong CD40 signaling induced by T_FH_ cells is required for development of CD80^hi^ B_mem_ cells”, second paragraph).

Finally, the authors look to mechanism and things become a little murky in so far as the confusion between in vitro activated and proliferating B cells and memory subsets remains. The conclusions that NFkB is involved and that its effects are mediated by IRF4 and BATF are to me highly problematic in themselves and also emblematic of my major issue with the work itself. These biochemistry studies, which also ignore a body of work on the consequences of CD40 signalling on GC B cells, are on actively proliferating B cells. One has to seriously question what the relationship is between a B cell cycling in optimal and unchanging conditions in vitro, to that in GC where presumably conditions and stimuli are under constant flux. Indeed, the authors show quite clearly that GC B cells are in fact CD80 negative while all their cells are CD80^+^. It would therefore seem that in vivo, B_mem_ formation is associated with acquisition of CD80 while in the in vitro transfer system, B_mem_ formation is the retention of CD80. Even in this context it is hard to understand how if NFkB is so crucial that knockdown of Rela has such a modest effect (Figure 5F), although the degree of knockdown is not shown.

We understand the reviewer’s concern, but we were aware that the mechanisms of CD40 signaling to induce CD80 expression in in vitro cultured B cells are primarily irrelevant to those to promote development of CD80^hi^ or CD80^lo^ B_mem_ cells in vivo. However, we thought that the biochemical analyses in the former could provide a clue to elucidate the latter mechanisms that appeared to be difficult to directly tackle. Thus, after having found that RelA was involved in the strong CD40 signaling in cultured B cells (the former Figure 5A, B), we tested whether RelA was involved in the development of CD80^hi^ B_mem_ cells (the former Figure 5E, F). The result showed that the RelA-knockdown in the in vivo activated B cells resulted in significant reduction of development of CD80^hi^ B_mem_ cells, although the difference was modest as the reviewer pointed out. However, the knockdown efficiency was unknown and could not be measured in this experiment because the ex vivo B cells transduced with the knockdown vector containing GFP marker had to be transferred back into mice 3 hours later, before the GFP expression became apparent (please see the new Figure 6G). Thus, we only tested the knockdown vector efficacy using iGB cells (the former Figure 5C). To convince the reviewer as well as the readers, in the revised manuscript, we have included the data of an experiment in which iGB cells were transduced with the knockdown vectors, now including that of c-Rel, and then transferred into mice to let them differentiate into memory-like iMB cells. As shown in the new Figure 6C-F and described in the Results section subsection “Possible mechanisms by which CD40 signaling facilitates GC B-cell differentiation into CD80^hi^ B_mem_ cells”, first paragraph) of the revised manuscript, the knockdown of either RelA or c-Rel, which was now evaluated (Figure 6C), significantly attenuated in vivo generation of CD80^hi^ iMB cells.

Although the reviewer seems to consider that a body of our work is on the consequences of CD40 signaling on GC B cells, we considered that the CD40 signaling affect B cells not only in the GC phase but also during the initial activation phase before the GC phase, for their differentiation into B_mem_ cells. Indeed, stimulation of naïve B cells for only two day, or even one, on the feeder cells differentially expressing CD40L affected their development into CD80^hi^ or CD80^lo^ iMB cells (the new Figure 3—figure supplement 1E-G). Thus, we consider that the iGB culture system represents the both phases, although being quite artificial, and is useful to elucidate signals for B cells to differentiate into B_mem_ cells (iMB cell in the experiments). Although majority of GC B cells are CD80^–^, unlike in vitro activated B cells, a minor fraction that has been selected for B_mem_ cells (pre-B_mem_ cells) expressed CD80 (the former Figure 6F, G; the revised Figure 7F, G). Therefore, the mechanism of CD40 signaling for the B_mem_ cell development may be similar between naïve B cells in the early phase and GC B cells in the GC phase of the primary immune response, and they both induce CD80 expression upon the strong and durable CD40 signaling and may retain the CD80 expression until and after they become B_mem_ cells. We have stated this view in the Discussion section of the new manuscript (fifth paragraph).

The final figure showing increased expression of CD80 on what are labelled memory precursors, together with increased IRF4 and BATF, is somewhat unconvincing as it is not clear if this amount of CD80 is the baseline for memory B cells, let alone if the slightly increased amounts of IRF4 and BATF are biologically relevant.

To answer this comment, we have repeated the same experiment but now including B_mem_ cells in the analysis. The expression levels of CD80 and BATF in pre-B_mem_ cells were intermediate between those of GC B cells and B_mem_ cells, although the increase of IRF4 expression in pre-B_mem_ cells was not significant in this experiment (the new Figure 7F, G). The data were evaluated by control staining with isotype-matched antibodies (the new Figure 7—figure supplement 1D). Thus, we revised the Results section in the revised manuscript (subsection “Possible mechanisms by which CD40 signaling facilitates GC B-cell differentiation into CD80^hi^ B_mem_ cells”, third paragraph).

My major concern is that this study represents two potentially independent and unrelated observations that are joined by the coincidence of CD80 expression. That is, why should I think that the processes occurring in the in vitro cultures reflect the processes in GC that give rise to memory? In vitro, the B cells are exposed to a constant and unlimited amounts of stimulus, whereas in GC they cycle between exposure to an unknown amount of CD40L, cytokines and cell-cell contact. Equally, IL21 is a crucial component of normal GC function and a defining characteristic of T_FH_, yet it is left out of these cultures. Previously this group reported that the B_mem_ produced by in vivo transfer showed different differentiation potential based on IL21 presence in the first culture. That confuses the CD40L as major factor quite a bit.

As answered in the previous response, we think the iGB culture system, from which the cultured B cells differentiate into B_mem_-like iMB cells in vivo, represents a common aspect of T-cell-mediated stimulation, but not the whole process, of naïve B cells and GC B cells during the initial phase of the immune response and the GC phase, respectively. The resultant CD80^hi^ and CD80^lo^ iMB cells and genuine CD80^hi^ and CD80^lo^ B_mem_ cells do not just coincide with each other on the CD80 expression, but were similar in their secondary responses in vitro as well as in vivo (Figure 1B, the new Figure 4, and Zuccarino-Catania et al., 2014). As the reviewer pointed out, the condition of the iGB culture system is far simpler than that of genuine GC, but it is well known that a numerous number of such simplified experimental systems have greatly contributed to our understanding of the complex immune system.

As for the IL-21, we previously reported that the addition of IL-21 secondary to the IL-4 in the iGB culture suppresses the generation of the iMB cells but instead promotes the generation of bone-marrow PCs (Nojima et al., 2011). Although we used only the first culture with IL-4 that allows generation of iMB cells in the original manuscript, we have now tested the effect of additional IL-21 in the first culture. As shown in the new Figure 3—figure supplement 1H, I (subsection “CD40 signal strength in vitro affects differentiation into CD80^hi^ or CD80^lo^ B_mem_ cells in vivo”, third paragraph) in the revised manuscript, the addition of IL-21 did not affect the frequency of CD80^hi^ iMB cells generated from iGB cells cultured on either 40LB^hi^ or 40LB^lo^ cells.

Similarly, what is the timing of CD80 upregulation in the cultures? Is it immediate and irrelevant to time or is it a timing event? What is the extent of proliferation in the different cultures? Is this phenomenon related to extent of division and would different results be achieved by transferring earlier?

We showed that CD80 was upregulated later than one day after the start of the iGB culture, and increased by day 3, to different extents depending on CD40L expression levels on feeder cells (Figure 3B). In our revised manuscript, we have included the data showing that the proliferation of iGB cells was equally minimum on day 2 but markedly differed depending on the feeder cells, as evident on day 4 after the start of the culture (the new Figure 3—figure supplement 1C). Transferring with the day 2 iGB cells, or even day 1, resulted in the similar result as with day 4 iGB cells, although the frequency of CD80^hi^ iMB cells corresponded to the length of the culture period, as shown in the new Figure 3—figure supplement 1E-G. This result indicates that the differential B_mem_ cell development is not related to the extent of division of B cells in culture, but related to strength and duration of CD40 stimulation. These results and view are stated in the Results section of the revised manuscript (subsection “CD40 signal strength in vitro affects differentiation into CD80^hi^ or CD80^lo^ B_mem_ cells in vivo”, third paragraph).

Last why are all the mice examined at such early time points? What kind of memory is present at days 7 or 10, and doesn't that change dramatically over the weeks that follow? As shown in this reports Figure 2?

B_mem_ cells were examined at 6 weeks after immunization in the experiment shown in Figure 1C and Figure 2B, but mainly at 10 days (or 7 days in some) in other experiments. This is because we focused on the generation of B_mem_ cells and tried to avoid outcomes of possible alterations in the maintenance of B_mem_ cells or in their late development from GC, which might be caused by the experimental interventions we performed. The previous reports indicated that class-switched B_mem_ cells are readily detectable by day 7~9 in experiments where antigen-specific Ig-knockin B cells are transferred into mice, and most B_mem_ cells are formed by day 11 after immunization (Wang et al., 2017; Suan et al., 2017; Weisel et al., 2016). We stated the above view in the Results section of the revised manuscript (subsection “Strong CD40 signaling induced by T_FH_ cells is required for development of CD80^hi^ B_mem_ cells”, third paragraph).

Reviewer #3:In this manuscript Koike and colleagues investigate the role of CD40-CD40L interaction for the development of functional different memory B cell (MBC) subsets i.e. MBCs that upon re-activation are mostly prone to differentiate into plasma cells (CD80^+^) and those (CD80-) mostly prone to re-enter the germinal center (GC) reaction. In light of these functional properties, and in parallel with terms used for T cells, the authors respectively call these subsets effector memory B (B_EM_) and central memory B (B_CM_). Notably, a productive GC reaction is largely required for the formation of the so-called B_EM_, while a GC is largely dispensable for the formation of B_CM_. Overall the study demonstrates that CD40 is required for the formation of MBC cell subsets characterized by CD80^+^ or CD80-. The authors also show that B_EM_ formation requires increased CD40 signaling compared to B_CM_ and this correlates to their GC and pre-GC formation, respectively. BCR affinity contributes to the ability of cells to present antigen and therefore receive increased help via CD40, as previously shown by others. The work also shows that once B_EM_ and B_CM_ are formed their functional properties are per se largely independent on BCR receptor signal strength, suggesting a different transcriptomic or epigenetic profile. MBCs displaying IgG1 positivity can also be formed in the absence of a GC reaction, and the authors do not perform experiments where somatic mutation is determined. Does it remain to be shown whether in the GC context the choice of a B cell to differentiate into B_EM_ or B_CM_ exists and if that choice is dependent on CD40 signaling strength. There are additional conceptual and technical aspects of the work that need to be addressed.

Before responding to each comment, we note here that we have got rid of the nomenclature, B_EM_ and B_CM_, and instead called them as CD80^hi^ and CD80^lo^ B_mem_ cells throughout our revised manuscript, in response to other reviewers’ and editors’ comments.

Our response to the sentence “*Does it remain to be shown whether in the GC context the choice of a B cell to differentiate into B_EM_ or B_CM_ exists and if that choice is dependent on CD40 signaling strength.*”:

Although we have not done such experiments, Shlomchik and colleagues showed in their paper that injection of blocking anti-CD40L antibody at the peak of GC reaction (day 12-14 after immunization) resulted in reduction of the frequency of CD80^+^ B_mem_ cells and increase of CD80^–^ B_mem_ cells, suggesting that the choice of a B cell to differentiate into either B_mem_ cell type, depending on CD40 signaling strength, exists in the GC context (Weisel et al., 2016).

1) Figure 1B (Figure 1—figure supplement 1) – The Flow Cytometry gating strategy in Figure 1—figure supplement 1B that pertains the results summarized in Figure 1B needs to be revisited. What is the justification for inclusion of GL7 within the plasma cell gating? Also, GL7 expression within the CD138- subset displays a broad range of intensities and the gating in this case includes GL7 low and likely also GL7 negative cells. How would the results in Figure 1B appear if the gating strategy is more conservative? Staining for Bcl6, as a GC marker, and for Blimp1, as a plasma cell marker, would clarify this issue.

We have re-gated the FCM data by quadrants (the new Figure 1—figure supplement 1B) and re-assessed the frequencies of CD138^+^ GL7− plasmablasts and CD138− GL7^+^ GC B cells (the new Figure 1B), which resulted in the similar data as the former Figure 1B.

2) Figure 2B – Transfer of B1-8 B cells into CD40lg^-/-^ yields very few cells with an MBC phenotype demonstrating that CD40 engagement is largely required for MBC formation in general. How much or how little do the few formed MBC cells in CD40lg^-/-^ express CD80?

The number of the B_mem_ cells in CD40lg^–/–^ mice was so few in some individuals that we could not make statistically reliable data. If we took data from a few CD40lg^–/–^ mice with relatively higher number of B_mem_ cells, the frequency of CD80^hi^ B_mem_ cells was decreased by about half as compared to those in CD40lg^+/+^ mice (data not shown).

3) Figure 2C, D – What is the impact in GC B cell frequency and numbers and how does that confound the results presented and interpretation?

In the mice injected with anti-CD40L antibody, the frequency and the numbers of GC B cells declined by about one-tenth as deduced by the data in Figure 2C, and as shown in the new Figure 2—figure supplement 1E, respectively. Therefore, it is unclear whether the reduction of CD80^hi^ B_mem_ cells was due to the decrease of GC B cells or to the reduction of CD40 signaling in GC B cells, or both. That is why we used the iGB culture system with differential in vitro CD40 stimulation in Figure 3.

What is the authors interpretation for the cell number recovery of both B_EM_ and B_CM_ by 6 weeks?

We think this is because CD80^hi^ B_mem_ cells may have been generated in later GC after the injected anti-CD40L antibody had lapsed, as we have described in the Results section of the revised manuscript (subsection “Strong CD40 signaling induced by T_FH_ cells is required for development of CD80^hi^ B_mem_ cells”, third paragraph).

4) Figure 3 – How do the varied levels of CD40L expression (40LB-lo, mid, hi) in the 40LB culture system impact on the generation of iGB cells in vitro? What is the phenotype, beyond CD80 expression, of the cells to be transferred into mice? As mentioned previously the study would have increased credibility and interpretations could be made with higher confidence if throughout the study Bcl6, as a GC marker, and for Blimp1, as a plasma cell marker are used.

In response to this comment, we analyzed surface phenotype, proliferation, and expression of *Bcl6* and *Prdm1* (for Blimp1) mRNA (the new Figure 3—figure supplement 1A-D). The extent of class switching and expression levels of GL7, PD-L2, CD73 were almost the same among the B cells cultured on 40LB-lo, 40LB-mid, or 40LB-hi. Expression levels of CD38 and CD62L were higher and those of Fas were lower in B cells cultured on 40LB-lo feeder cells, probably reflecting less activated state of the cells. Proliferation was correlated to the expression levels of CD40L on these feeder cells. Curiously, the mRNA levels of both *Bcl6* and *Prdm1* exhibited inverse correlation to the CD40L levels on these feeder cells. We described about these data in the Results section of the revised manuscript (subsection “CD40 signal strength in vitro affects differentiation into CD80^hi^ or CD80^lo^ B_mem_ cells in vivo”, first paragraph).

5) Figure 5E – Also in this experiment it is important to describe the phenotype of the transferred cells.

In this experiment (the new Figure 6G, H), the B cells activated in vivo, and transduced with the knockdown or control vectors containing GFP marker ex vivo, had to be transferred back into mice 3 hours after the gene transduction, before the GFP expression became apparent (the new Figure 6G). Since we could not distinguish the vector-transduced (GFP^+^) cells before transfer, we could not assess the phenotypic change by the gene knockdown.

To compensate for the weakness of this experiment, we performed an experiment in which iGB cells were transduced with the knockdown vectors, now including that of c-Rel, and then transferred into mice to let them differentiate into memory-like iMB cells. The knockdown of RelA or c-Rel did not affect class-switching and did not induce PC differentiation of iGB cells before the transfer (the new Figure 6—figure supplement 1A), but significantly attenuated in vivo generation of CD80^hi^ iMB cells, as shown in the new Figure 6C-F and described in the Results section (subsection “Possible mechanisms by which CD40 signaling facilitates GC B-cell differentiation into CD80^hi^ B_mem_ cells”, first paragraph) of the revised manuscript.

Upon transfer what is the impact of RelA knockdown on B_CM_, the GC reaction, plasma cell formation?

We have now included data showing the frequency of GC B cells among the gene-transduced, antigen-specific, class-switched B cells, and CD80^lo^ cells among the gene-transduced B_mem_ cells, as shown in the new Figure 6—figure supplement 1C in the revised manuscript. PCs could not be seen in the spleen at this time point. The RelA knockdown resulted in a little increase of the frequency of GC B cells and increase of CD80^lo^ B_mem_ cells complementary to the decrease of CD80^hi^ B_mem_ cells.

6) Figure 6. What would be the outcome in the formation of either B_EM_ and B_CM_ upon transfer of iGC-lo cells constitutively or conditionally expressing BATF, IRF4 or both?

We have performed this experiment, as shown in the new Figure 7H-J in the revised manuscript. The result demonstrated that BATF-expressing (GFP^+^) iMB cells contained a significantly higher proportion of CD80^hi^ cells than mock-transduced iMB cells. IRF4-expressing (CFP^+^) cells could not be detected in iMB cells, possibly because those over-expressing IRF4 had differentiated into PCs. We described about this data in the Results section of the revised manuscript (subsection “Possible mechanisms by which CD40 signaling facilitates GC B-cell differentiation into CD80^hi^ B_mem_ cells”, last paragraph).

7) The authors propose, in the fourth paragraph of the Discussion, that the reason why the so-called B_CM_ cells are hardly generated during the GC reaction is because a) these cells fail to induce a high level of CD40L in the cognate T cells and hence would die by apoptosis. b) The authors also write that apoptosis can be avoided by c-Myc expression downstream of CD40 signaling. While the first point (a) is likely to be true given the survival properties of CD40 signaling, the suggestion that c-Myc curtails apoptosis downstream of CD40 (b) is an experimentally unsubstantiated assumption. Current data suggests that c-Myc positivity in the GC reaction represent positive selection downstream of BCR and CD40 signaling and that is most likely the reason why these cells display reduced apoptosis compared to c-Myc negative LZ B cells.

In accordance with the reviewer’s comment, we have removed the statement of our assumption about c-Myc involvement from the Discussion section.

8) The authors go through great lengths to discredit a fundamental role for CD40 in plasma cell differentiation, suggesting that CD40 signaling merely enhances the effect of cytokine from particular T_FH_ cells, likely IL21 expression ones. Clearly this is a gross misinterpretation, previous work has demonstrated that ablation of RelA is critical for GC derived plasma cell formation. Having said this, CD40 signaling may not be per se sufficient; as the authors themselves write IL21 has also been shown to be required for plasma cell formation to ensue. It is not contradictory that CD40 is involved in both MBC and plasma cell formation given that the cell fate is likely to be ultimately determined by a combination and integration of signaling pathways.

We agree to this comment, and therefore we have removed the statement that the strong CD40 signaling merely enhance the interaction between GC B cells and T_FH_ cells from our Discussion, and extensively revised this paragraph of the Discussion in the revised manuscript, adapting the reviewer’s view, namely, a combination and integration of CD40 signaling and cytokine signaling may determine the fate of GC B cells (Discussion, sixth paragraph).

9) The authors propose that modulation of CD40 signaling could facilitate the generation of B_CM_ in vaccination possibly beneficial against evolved viruses with epitope mutations. This proposal is however negated by the authors data in Figure 2: if CD40 signaling is impaired (Figure 2C) the formation of both B_EM_ or B_CM_ is impaired (cell numbers); if CD40 signaling is enhanced (Figure 2E) there is no impact on the production of B_CM_ (cell numbers), but a boost occurs in the production of B_EM_ (cell numbers).

We completely agree to this comment and have revised our Discussion in our revised manuscript as following: “Thus, if we could dissect the CD40 signaling pathways that direct differentiation into CD80^hi^ B_mem_ cells and those promote cell proliferation/survival, selective suppression of the former pathways during vaccination might convert the generation of CD80^hi^ B_mem_ cells into CD80^lo^ B_mem_ cells that would eventually produce broadly reactive Abs.”.

An avenue with respect to skewing the B_EM_ GC fate to a B_CM_ one may be possible, however as mentioned above, the authors do not perform experiments addressing whether in the GC context a B cell fate choice to B_EM_ or B_CM_ exists and if that choice is dependent on CD40 signaling strength.

As for this part of the comment, we have described our response following the “General assessment and major comments”.